**Total Atmospheric Mercury Deposition in Forested Areas in Korea**
Jin-Su Han[1], Yong-Seok Seo[1,2], Moon-Kyung Kim[1,2], Thomas M. Holsen[3], Seung-Muk Yi[1,2,*]
[1] Department of Environmental Health Sciences, Graduate School of Public Health, Seoul
National University, 1 Gwanak-ro, Gwanak-gu, Seoul 08826, South Korea
[2] Institute of Health and Environment, Seoul National University, 1 Gwanak-ro, Gwanak-gu,
Seoul 08826, South Korea
[3] Department of Civil and Environmental Engineering, Clarkson University, Potsdam,
NY13699, USA
*Address correspondence to Dr. Seung-Muk Yi, Graduate School of Public Health, Seoul
National University, 1 Gwanak-ro, Gwanak-gu, Seoul 088626, South Korea
E-mail) yiseung@snu.ac.kr
Telephone) 82-2-880-2736
Fax) 82-2-745-9104



**Abstract**
In this study, mercury (Hg) was sampled weekly in dry and wet deposition and
throughfall and monthly in litterfall, and as it was volatilized from soil from August 2008 to
February 2010 to identify the factors influencing the amount of atmospheric Hg deposited to
forested areas in a temperate deciduous forest in Korea. For this location there was no
significant correlation between the estimated monthly dry deposition flux (litterfall +
throughfall – wet deposition) (6.7 $\mu g$ $m^{-2}$ $yr^{-1}$) and directly measured dry deposition (9.9 $\mu g$
$m^{-2}$ $yr^{-1}$) likely due primarily to Hg losses from the litterfall collector. Dry deposition fluxes
in cold seasons (fall and winter) were lower than in warmer seasons (spring and summer).
The volume-weighted mean (VWM) Hg concentrations in both precipitation and throughfall
were highest in winter likely due to increased scavenging by snow events. Since Korea
experiences abundant rainfall in summer, VWM Hg concentrations in summer were lower
than in other seasons. Litterfall fluxes were highest in the late fall to early winter when leaves
were dropped from the trees (September to November). The cumulative annual Hg emission
flux from soil was 6.8 $\mu g$ $m^{-2}$ $yr^{-1}$. Based on these data, the yearly deposition fluxes of Hg
calculated using two input approaches (wet deposition + dry deposition or throughfall +
litterfall) were 6.8 and 3.6 $\mu g$ $m^{-2}$ $yr^{-1}$ respectively. This is the first reported study which
measured the amount of atmospheric Hg deposited to forested areas in Korea and thus our
results provide useful information to compare against data related to Hg fate and transport in
this part of the world.


**Keywords**: Mercury budget; Dry deposition; Wet deposition; Throughfall; Litterfall; Hg
emission flux

## 1. Introduction

Mercury (Hg) is a highly toxic pollutant and a threat to human health and ecosystems due to its ability to bioaccumulate and biomagnify through the food chain after it is methylated (Lindqvist et al., 1991; Schroeder and Munthe, 1998). It is classified as a persistent bioaccumulative and toxic (PBT) chemical (U.S.EPA, 1997a). Atmospheric Hg exists in three different forms with different chemical and physical properties; gaseous elemental mercury (GEM, $Hg^0$), gaseous oxidized mercury (GOM, $Hg^{2+}$), and particulate bound mercury (PBM, $Hg_p$). GEM is the major form of Hg in the atmosphere and is relatively water insoluble and very stable with a long residence time of 0.5 - 2 years (Carpi and Lindberg, 1997; Cohen et al., 2004; Schroeder and Munthe, 1998; Zhu et al., 2014). GOM is water soluble, with relatively strong adhesion properties (Han et al., 2005) and can be scavenged by rain within precipitating and below clouds (Blackwell and Driscoll, 2015; Lin and Pehkonen, 1999). The dry deposition velocity is similar to $HNO_3$ (1~5 cm sec$^{-1}$) if it is assumed that all GOM is in the form of $HgCl_2$ (Petersen et al., 1995). PBM is formed by GEM or GOM adsorbing to a particle (Lai et al., 2011). Atmospheric PBM transport is significantly affected by its particle size distribution and may contribute to both wet and dry deposition (Lynam and Keeler, 2002).

Wet and dry deposition of atmospheric Hg is an important input to the aquatic and terrestrial ecosystems (Buehler and Hites, 2002; Fitzgerald et al., 1998; Landis and Keeler, 2002; Lindberg et al., 1998; Miller et al., 2005; Rolfhus et al., 2003; Selvendiran et al., 2008; Shanley et al., 2015). Hg deposited from the atmosphere can be transformed to methyl mercury (MeHg) which bio-accumulates in aquatic food chains, resulting in adverse health and ecological effects (Ma et al., 2013; Lindberg et al., 2007; Rolfhus et al., 2003; Selin et al., 2007; Weiss-Penzias et al., 2016; Zhu et al., 2014). Atmospheric Hg deposition to forests includes direct dry deposition, throughfall, and litterfall. Dry deposition to leaves compromises a large proportion of litterfall (Grigal, 2002; St. Louis et al., 2001). Previous investigations (Fu et al., 2009) estimated dry deposition to forested areas as litterfall + throughfall – wet deposition. However, there are many variables that can adversely influence this technique including reemitted Hg from beneath the canopy and sampling artifacts. Directly measuring dry deposition with a surrogate surface is an alternative approach, although there is no universally accepted method on how to make these measurements.

Hg deposited onto plant surfaces can be revolatilized, incorporated into tissue or
washed off by precipitation (which is deemed throughfall) which often results in throughfall
having higher Hg concentrations than precipitation (Iverfeldt, 1991; Kolka et al., 1999;
Munthe et al., 1995; Choi et al., 2008; Grigal et al., 2000; Schwesig and Matzner, 2000).
Litterfall is dead plant material such as leaves, bark, needles and twigs that has fallen to the
ground. Litterfall carries new Hg inputs from the atmosphere to the forest floor and also Hg
recycled from volatilization from soils and other surfaces. Throughfall and litterfall
contribute to the biochemical recycling of atmospheric Hg in forest systems (St. Louis et al.,
2001) and are important Hg inputs that result in Hg accumulation in forest systems
(Blackwell and Driscoll, 2015).
The deposition of Hg in the forest ecosystem is complicated because of complex
interactions between atmospheric Hg and the canopy, including oxidation of Hg on leaf
surfaces (Blackwell and Driscoll, 2015; Iverfeldt, 1991), deposition of GOM and PBM on
leaf surfaces (Blackwell and Driscoll, 2015; Blackwell et al., 2014; St. Louis et al., 2001),
stomatal uptake of atmospheric GEM (Fu et al., 2010; Iverfeldt, 1991; Lindberg et al., 1991;
St. Louis et al., 2001), root uptake of dissolved Hg in soil and soil water and stomatal uptake
of GEM that was volatilized from soils (Bishop et al., 1998; Cocking et al., 1995; Ma et al.,
2015; St. Louis et al., 2001). Also, the Hg in forest canopies can be emitted and reemitted
from beneath the canopy (Risch et al., 2012). The Hg mass in litterfall have orginated from a
large portion of dry deposition (Risch et al., 2012; St. Louis et al., 2001).
To date there have been few studies (Blackwell et al., 2014; Choi et al., 2008; Rea et
al., 2001) that have estimated atmospheric Hg deposition to forested areas and none in Korea.
Fully characterizing Hg deposition in forested areas is important for estimating
environmental risks associated with Hg. Thus, the objectives of this study were to
characterize total atmospheric Hg deposition in a temperate deciduous forested area in Korea
by measuring Hg dry deposition, wet deposition, throughfall, litterfall and volatilization from
soils and comparing directly measured and estimated dry deposition. Based on the collected
data the annual Hg flux was estimated using two approaches to determine inputs (wet
deposition + dry deposition, throughfall + litterfall) minus volatilization from soil.

**2. Materials and methods**

**2.1. Site description**

The sampling sites were located at Yangsuri, Yangpyeong-gun, Gyeonggi-do, a

province in Korea where the Bukhan (North Han) and Namhan River (South Han River)
come together (Fig. 1). Gyeonggi-do has a population of 12 million (24% of the total
population and the most populated province in South Korea) and an area of 10,187 $km^2$ (10%
of the total area of South Korea). Yangpyeong-gun has a population of 83,000 (0.2% of the
total population in South Korea) and an area of 878.2 $km^2$ (0.9% of the total area in South
Korea). Wet deposition samples were collected at the Han River Environment Research
Center (Elevation 25 m, N37°32´, E127°18´) (site A in Fig. 1). Dry deposition, throughfall,
litterfall, volatilization from soils and total mercury (TM) in soil samples were determined in
a deciduous forest including primarily chestnut (Elevation 60 m, N37°32´, E127°20´) (site B
in Fig. 1) about 2 km away from site A. This area contains rivers, a flood plain, agricultural
land, residential areas, forests, and wetlands. Therefore, the study sites are appropriate for
identifying the in/out flow of Hg in a forested ecosystem typical for this part of the world.

**2.2. Sampling methods**

Samples were collected from August 2008 to February 2010. Weekly samples for dry

and wet deposition in an open area and throughfall were collected using a dry and wet
deposition sampler (DWDS).

2.2.1. Dry deposition for GOM and PBM

Some studies have investigated the use of surrogate surfaces to directly measure Hg

dry deposition (Lyman et al., 2007; Peterson and Gustin, 2008). Surrogate surfaces allow
better control over exposure times than those provided with natural vegetation (Lai et al.,
2011). However, surrogate surfaces, being smooth, may not mimic Hg dry deposition to
natural rougher surfaces (Huang et al., 2011). Surrogate surfaces with cation exchange
membranes have been useful for measuring GOM however they may collect a very small
aerosol fraction by diffusion (Huang and Gustin, 2015; Lyman et al., 2007). Similar to
previous studies, in this project the dry deposition sampler was equipped with a knife-edge
surrogate surface (KSS) sampler with the collection media facing up. Forty seven-mm quartz
filters were used to measure PBM deposition and KCl-coated quartz filters were used to
measure GOM + PBM deposition. The quartz filter and KCl-coated quartz filter (soaked in
KCl solution for 12h and dried on clean bench) were pre-baked in a quartz container at 900
ºC for PBM and 525 ºC for GOM + PBM. Before weekly sampling, the filters were placed on
a filter holder base and held in place with a retaining ring and then were placed on the KSS.
Filter exposed to the atmosphere from approximately one week and two side-by-side samples
were deployed during each dry day.

2.2.2. TM in wet deposition and throughfall
The DWDS for wet deposition and throughfall was equipped with four discrete
sampling systems that allows for two Hg and two trace elements sampling trains similar to
what was used in previous studies (Lai et al., 2007; Landis and Keeler, 1997; Seo et al., 2012;
Seo et al., 2015).

2.2.3. TM in soil and litterfall
Soil samples were collected every month from December 2008 to October 2010,
except January 2009, January, July, and August 2010, at depths of 6 (A horizons) and 15 cm
(B horizons).
Litterfall samples was collected every month from December 2008 to November
2010, except January 2010. Ten nylon-mesh-lined baskets (1.09 m$^2$ each) were acid cleaned
and randomly placed under the canopy. All litter and soil samples were freeze-dried, sorted
by tree species, weighed, and then homogenized by crushing manually prior to analysis.

2.2.4. Volatilization from soils
The gaseous mercury emission flux from soil was measured using a dynamic flux
chamber (DFC) connected to the Tekran 2537A (Tekran Inc., Toronto, Canada) and Tekran
1110 dual sampling unit (allows alternate sampling from inlet and outlet) (Choi and Holsen,
2009b) under the deciduous forest area once a month. Daily automated calibrations were
performed for the Tekran 2537A using an internal permeation source. Manual injections were
used to evaluate these calibrations using a saturated mercury vapor standard. The flowrate
was approximately 5 L min$^{-1}$. Four 1 cm diameter inlet holes were evenly placed around the
chamber ensuring it was well mixed. The bottom 2 cm of DFCs (3.78L) were covered by soil.
The DFCs were made of glass and polycarbonate which may block some UV light (Choi and
Holsen, 2009a; Skinner, 1998).

**2.3. Analytical methods**

2.3.1. Dry deposition for GOM and PBM

The dry deposition samples for GOM and PBM samples were analyzed using a tube

furnace connected to a Tekran 2537. The tube furnace was pre-heated (GOM: 525 °C, PBM:
900 °C) and zero air passed through until the Hg concentration was zero (Kim et al., 2009;
Kim et al., 2012). After samples were placed inside the tube furnace, the tube furnace was
purged with zero air until Hg level was again zero. The mass of Hg desorbed from the sample
was determined using the product of concentration and flowrate ($5 \text{ L min}^{-1}$). The system
recovery was measured by injecting mercury vapor standards (0, 10, 20, 30, 50 μL) manually.
It was assumed that GOM deposition was equal to the flux measured by the KCl-coated
quartz filter minus the flux measured by the quartz filter. However, recent studies (Lyman et
al., 2010) reported potential sampling artifacts in the presence of $O_3$.

2.3.2. TM in wet deposition and throughfall

TM in throughfall was measure using a Tekran Series 2600 equipped with cold vapor

atomic fluorescence spectrometer (CVAFS) following the procedures outlined in the U.S.
EPA Method 1631 version E (U.S.EPA, 2002) and the U.S. EPA Lake Michigan Mass
Balance Methods Compendium (LMMBMC) (U.S.EPA, 1997b)

2.3.3. TM in soil and litterfall

TM concentrations in soil and litterfall samples were determined using a direct

mercury analyzer (DMA-80, Milestone, Italy), which utilizes the serial process of thermal
composition, catalytic reduction, amalgamation, desorption, and atomic absorption
spectroscopy.

**2.4. QA/QC**

2.4.1 Dry deposition for GOM and PBM

Automated daily calibration of Tekran 2537A routinely was performed using an
internal permeation source. Two-point calibrations (zero and span) were performed
separately for each pure gold cartridge. A recovery of $102 \pm 2.9\%$ ($r^2 > 0.9995$) ($n = 4$) was
measured by directly injecting knowing amounts of five Hg standards which was connected
to zero air. The Method Detection Limit (MDL) determined by measuring the Hg
concentration in zero air was 0.04 ng $m^{-3}$. Additional information is provided in the SI.

2.4.2. TM in wet deposition and throughfall
Quality assurance and quality control were based on the U.S. EPA Methods 1631
version E (U.S.EPA, 2002) and LMMBMC (U.S.EPA, 1997b). The MDL (three times the
standard deviation of seven sequential reagent blanks) for TM in wet deposition and
throughfall was 0.05 ng $L^{-1}$. The standard curve was acceptable when $r^2$ was greater than
0.9995 (linear). More additional information is described SI.

2.4.3. TM in litterfall and soil
TM in litterfall and soil was reported on a dry-weight basis. Recovery (%) of
standard reference materials (SRMs) (MESS3, marine sediment) purchased form the National
Research Council of Canada and analyzed every 10 samples at the start of experiments was
$104 \pm 4\%$.

2.4.4. Volatilization from soil
The DFC was connected to the Tekran 2537A through Tekran 1110 sampling unit.
Ten $\mu$L of vapor phase Hg was injected into the DFC ($n = 10$) before deployment in the field.
Recovery was $86 \sim 110\%$ and averaged 101% at a flow rate of 5 L $min^{-1}$. Before flux
chamber measurements automated calibration was performed using the internal permeation
source connected to the Tekran 2537A and Tekran 1110 dual sampling unit. External
calibration and MDLs for this instrument are described above.


**3. Results and Discussion**

**3.1. Monthly and seasonal variations in dry deposition fluxes of GOM and PBM**

247   Weekly samples were collected using quartz filters (PBM) and KCl coated quartz

248 filters (GOM). The average dry deposition fluxes for GOM (Table S1) and PBM (Table S2)

249 were 5.4 µg m$^{-2}$ yr$^{-1}$ (range: 0.4 ~ 14.4 µg m$^{-2}$ yr$^{-1}$) and 4.3 µg m$^{-2}$ yr$^{-1}$ (range: 0.8 ~ 19.4 µg

250 m$^{-2}$ yr$^{-1}$), respectively. The dry deposition fluxes for GOM were highest in spring 2009 (10.0

251 ± 2.0 µg m$^{-2}$ yr$^{-1}$), lowest in fall 2009 (1.2 ± 1.4 µg m$^{-2}$ yr$^{-1}$) while the dry deposition fluxes

252 for PBM were highest in summer 2009 (9.6 ± 9.0 µg m$^{-2}$ yr$^{-1}$), lowest in fall 2009 (1.2 ± 0.4

253 µg m$^{-2}$ yr$^{-1}$) (Fig. 2). Nonparametric Mann-Whitney U tests indicated that there were

254 statistically significant differences in the dry deposition fluxes for GOM between spring 2009,

255 fall 2008, and fall 2009 ($p < 0.05$) and there were statistically significant differences in the

256 dry deposition flux for PBM between summer 2009 and fall 2009 ($p < 0.05$).

257   Zhang et al. (2012) reported that in eastern and central North America the GEM

258 concentration in the colder seasons were generally higher than in warmer seasons. However,

259 the dry deposition fluxes for GOM and PBM in spring and summer (warmer seasons) were

260 higher than in the fall and winter (cold seasons) following the same pattern as average GEM

261 concentrations (summer 2009: 2.7 ± 0.9 ng m$^{-3}$, spring 2009: 2.4 ± 0.6 ng m$^{-3}$, fall 2009: 2.3

262 ± 0.7 ng m$^{-3}$, winter 2008: 1.2 ± 0.2 ng m$^{-3}$) in Han River Environment Research Center

263 (located approximately 2 km away).

266 **3.2. Monthly and seasonal variations of TM wet deposition and throughfall flux**

267   The average VWM concentration in precipitation (n = 35) and throughfall (n = 44)

268 are shown Fig.3. Nonparametric Mann-Whitney U tests indicated that there were no

269 statistically significant differences in the VWM TM concentration between winter 2009 and

270 other seasons which is probably related with the small number of samples. The VWM TM

271 concentration in winter 2009 was statistically significantly higher than fall 2009 ($p = 0.007$),

272 spring 2009 ($p = 0.035$), and summer 2009 (p = 0.001) in throughfall.

273   The high VWM Hg concentrations in precipitation and throughfall in winter were

274 likely associated with the combined effects of reduced mixing heights (Blanchard et al.,

275 2002) which increases atmospheric concentrations (Kim et al., 2009; Seo et al., 2015), low

276 rainfall depth (11.7% of total rainfall depth) which is a typical pattern in Yangpyung, Korea

277 (KMA,

278 http://www.kma.go.kr/weather/climate/average_30years.jsp?yy_st&tnqh_x003D;2011&

stn&tnqh_x003D;108&norm&tnqh_x003D;M&obs&tnqh_x003D;0&mm&tn
qh_x003D;5&dd&tnqh_x003D;25&x&tnqh_x003D;25&y&tnqh_x003D;5
(accessed May 5, 2016) and the inclusion of snow events since scavenging by snow is more
efficient than by rain due to the larger surface area of snow (snow: 700 $cm^2$/g, rain: 60 $cm^2$/g)
(Kerbrat et al., 2008). While, Sigler et al. (2009) reported that GOM is scavenged less
efficiently during snow events.

Previous studies reported that rainfall depth in forested areas were approximately

8~24% smaller than that in an open area (Choi et al., 2008; Deguchi et al., 2006; Keim et al.,
2005; Price and Carlyle-Moses, 2003) due to capture by the foliage and subsequent
evaporation. In this study, rainfall depth in the forest was approximately 8% smaller than that
in the open area. Regression analysis revealed that the TM concentration in throughfall was
higher than in precipitation (statistically significant differences ($r^2 = 0.20$) ($p < 0.05$)) due to
wash off of previously deposited Hg from the foliage (Grigal et al., 2000; Iverfeldt, 1991;
Kolka et al., 1999; Schwesig and Matzner, 2000) and oxidation of $Hg^0$ to $Hg^{2+}$ on the wet
foliage surface by ozone and subsequent wash off (Graydon et al., 2008). Other possible
sources of Hg in throughfall are leaching and biogeochemical recycling of Hg from foliage
(St. Louis et al., 2001). Some of the deposited Hg can be washed off by rainfall and reemitted
as GEM to the atmosphere (Jiskra et al., 2015; Rea et al., 2001). Therefore, all of the Hg
deposited on the foliar surfaces is not in the throughfall. Throughfall also incorporates GOM
and PBM that is adsorbed from the atmosphere by leaves since GOM is soluble and it is
likely readily washed off during rain events (Blackwell and Driscoll, 2015).


**3.3 Relationship between rainfall depth, VWM TM concentration, TM wet deposition**
**and throughfall flux**

There was a statistically significant negative correlation between rainfall depth and

VWM TM concentrations in precipitation ($r^2 = 0.13$) ($p < 0.05$) (Fig. S1) and throughfall ($r^2 =$
0.19) ($p < 0.05$) (Fig. S2) due to dilution during the later stage of a precipitation event. This
negative correlation has also been found in previous studies (Guo et al., 2008; Landis and
Keeler, 2002; Seo et al., 2012; Seo et al., 2015; Wallschläger et al., 2000). About 19% of
throughfall and 13% of precipitation variation in VWM concentration are explained by
precipitation depth. The rest of the variation is likely due to meteorological parameters that
differ between events (Gratz et al., 2009), for example temperature (Table S3) and
precipitation type (Rain, Snow, Mixed) and variations in ambient Hg speciation and PBM
particle size distributions due to differing impacts of local and regional sources (Blackwell
and Driscoll, 2015). There was a statistically significant positive correlation between rainfall
depth and TM deposition flux in precipitation ($r^2 = 0.34$) ($p < 0.05$), similar to what was
found in previous studies (Choi et al., 2008; Gratz et al., 2009; Shanley et al., 2015; Wang et
al., 2014), suggesting that the TM deposition flux increased during large events even though
continuous rain diluted the TM mass. However, a large rainfall depth does not affect wet
deposition fluxes significantly if GOM and PBM concentrations are low (Zhang et al., 2012).


**3.4. Leaf-on vs. Leaf-off**

At this sampling site the leaf-on season is from March to the end of November.
During leaf-on periods, the TM concentrations in throughfall (average 8.1 ng $L^{-1}$) were higher
than that in precipitation (average 5.4 ng $L^{-1}$) and regression analysis suggested that they
were significantly correlated ($r^2 = 0.59$) ($p < 0.05$). For leaf-off periods TM concentrations in
throughfall (average 14.3 ng $L^{-1}$) were 1.7 times higher than in precipitation (average 8.6 ng
$L^{-1}$) and concentrations were also significantly correlated ($r^2 = 0.56$) ($p < 0.05$) (Table 1). The
concentration enhancement during leaf-off periods was probably due, at least in part, to snow
on the branches that collected mercury due to dry deposition during dry periods that was
subsequently collected by the sampler after being blown off by wind and/or after it melted.
The sample-by-sample flux of Hg in throughfall was similar to or lower than that of
precipitation although the TM concentration in throughfall was higher than that in
precipitation. However the cumulative Hg fluxes in throughfall (leaf on: 7.0 µ$g$ Hg m$^{-2}$, leaf
off: 3.1 µ$g$ Hg m$^{-2}$) were higher than in precipitation (leaf on: 4.9 µ$g$ Hg m$^{-2}$, leaf off: 0.6 µ$g$
Hg m$^{-2}$). As mentioned previously this may be a result of differences in rainfall depth (leaf-on
periods) and snow events (leaf-off periods).


**3.5. TM in litterfall and soil**

Litterfall can be an important Hg input to soils under forested landscapes. The mean

monthly TM concentrations in litterfall were $50.2 \pm 16.5$ ng g$^{-1}$ (ranged from 28.2 to 76.4 ng
g$^{-1}$) (Fig. 4). TM litterfall fluxes from winter 2009 to fall 2010 (one year) were $0.3 \pm 0.4$ μg
m$^{-2}$ (ranged from 0.01 to 1.9 μg m$^{-2}$). TM litterfall fluxes varied depending on the sampling
periods; being lowest in summer, from June to August, and highest in fall, from September to
November (Fig. 4) because litterfall production increases substantially over the growing
season, from late fall to early winter. Hall and St. Louis (2004) reported the mean
concentration of TM in leaf litter increased from 7.1 ng g$^{-1}$ to a final value of 40.9 ng g$^{-1}$ in
deciduous litter. Demers et al. (2007) reported that the quantity of TM added to the decaying
deciduous leaf litter was 5.1 ~ 5.5 μg m$^{-2}$ during the growing season. In this study, TM
litterfall fluxes were smaller than those in previous studies.

Soil samples were collected from the near-surface A-horizon following the removal

of any rock fragments and the B-horizon. The mean soil TM concentrations were higher
within the A-horizon ($66.9 \pm 20.3$ ng g$^{-1}$) than within the B-horizon ($46.1 \pm 17.5$ ng g$^{-1}$). TM
concentration in soil collected in this study was similar to TM concentration found in soil
collected from uncontaminated baseline sites which ranged from 30 to 50 ng g$^{-1}$ (Gray et al.,

2015).



**3.6. Volatilization from soils**

Hg emission fluxes were estimated from directly measured soil volatilization of

gaseous elemental mercury (GEM) using a dynamic flux chamber (DFC). The measured
fluxes were the highest in June and the lowest in November. Emission fluxes were positively
correlated with ambient air temperature however, they were not influenced by precipitation.
For example, the ambient air temperature was higher in summer than other seasons, but were
not higher in July, a period of several severe rain storms nor were they lower in August which
had very little rain. This result may be because the relative humidity was high enough that the
soil remained moist. This result is similar to previous studies that found that Hg emission
fluxes were positively correlated with soil surface temperature and negatively correlated with
humidity (Choi and Holsen, 2009b; Gabriel et al., 2006; Wallschläger et al., 2000; Wang et al.,
2005). Hg emission fluxes during leaf-on periods (March to November) ($0.65 \pm 2.25$ ng m$^{-2}$
hr$^{-1}$,16.9 °C) were higher than leaf-off periods (December) ($0.02 \pm 2.13$ ng m$^{-2}$ hr$^{-1}$, -1.29 °C).
This result is similar to a previous study. Choi and Holsen (2009b) reported that during leaf-
off periods, the Hg emission flux was correlated with temperature and solar radiation. The
cumulative annual Hg emission flux was 6.8 $\mu g\ m^{-2}\ yr^{-1}$ (Fig. 5). Due to sampler (Tekran
2537A) malfunctions in January, February and April, fluxes were assumed to be equal to the
average of the flux of the previous and subsequent month. If only one month of data were
available, it was assumed to be the same as the missing month. For comparison the annual Hg
emission flux would be 4.8 $\mu g\ m^{-2}\ y^{-1}$ if only measured data were used.


**3.7 Estimated dry deposition at forest**

Fu et al. (2009) estimated dry deposition to be equal to litterfall + throughfall − wet

deposition. Using the data presented here, the estimated dry deposition flux (6.7 $\mu g\ m^{-2}\ yr^{-1}$)
was lower than measured dry deposition (9.9 $\mu g\ m^{-2}\ yr^{-1}$) and there was no significant
correlation between the two methods ($r^2 = 0.22$) ($p = 0.65$). One of the reasons for the directly
measured flux to be larger than the estimated flux is likely because there is no canopy
resistance for, or re-emission from, the KCl coated surrogate surface. The differences in the
estimates could be due to the loss of litter samples by wind or Hg losses from the collected
litter due to meteorological conditions such as rainfall (Blackwell et al., 2014) due to
relatively long sampling periods (1 month). However dry deposition collected with a
surrogate surface doesn't include accumulation in leaf stomata which may underestimate dry
deposition using this technique and since it is a smooth surface may collect less deposition
than a rougher surface.

The annual input flux calculated by summing wet deposition plus measured dry

deposition (14.2 $\mu g\ m^{-2}\ yr^{-1}$) was higher than the input flux calculated by summing
throughfall + litterfall (11.0 $\mu g\ m^{-2}\ yr^{-1}$) (Fig. 6). This difference is likely, at least in part, due
to the fact that no Hg is reemitted from wet and dry deposition as happens for litterfall.
Nonparametric Mann-Whitney U tests indicated that there were not statistically significant
differences ($r^2 = 0.14$) ($p = 0.98$). In general, wet + dry deposition was larger than throughfall
plus litterfall except during fall when leaves were being actively dropped from the trees. The
largest difference was in July during a period of significant precipitation (about 26.3 % of the
total amount in 2009). This difference is most likely due to the many reactions and
transformations on the leaf surface that aren't mimicked with the surrogate surface including
re-emission (Rea et al., 2001).


**3.8. Mercury budget**


The yearly estimated budget of Hg in this study site was calculated using both input

approaches (Total input = wet deposition + dry deposition or Total input = throughfall +
litterfall) as follows. 1) Input to the forest canopy (wet deposition in an open area: 4.3 $\mu$g m$^{-2}$
yr$^{-1}$ plus dry deposition in the forested area: 9.9 $\mu$g m$^{-2}$ yr$^{-1}$) minus output (emissions from
soil 6.8 $\mu$g m$^{-2}$ yr$^{-1}$ plus accumulation in the soil 0.6 $\mu$g m$^{-2}$ yr$^{-1}$) resulting in a net flux of 6.8
$\mu$g m$^{-2}$ yr$^{-1}$. 2) The alternative method yields input (throughfall: 6.4 $\mu$g m$^{-2}$ yr$^{-1}$ plus litterfall:
4.6 $\mu$g m$^{-2}$ yr$^{-1}$) minus output (emissions from soil: 6.8 $\mu$g m$^{-2}$ yr$^{-1}$ plus accumulation in the
soil: 0.6 $\mu$g m$^{-2}$ yr$^{-1}$) resulting in a net flux of 3.6 $\mu$g m$^{-2}$ yr$^{-1}$. For comparison at the
Lehstenbach catchment in Germany, the estimated net fluxes were similar: 6.8 $\mu$g m$^{-2}$ yr$^{-1}$
(Schwesig and Matzner, 2000) and in the Experimental Lakes Area (ELA) watersheds in
Canada, the flux was 3 ~ 4 $\mu$g m$^{-2}$ yr$^{-1}$ (St. Louis et al., 2001). However, for the Lake
Langtjern spruce forest in southeast Norway (20.1 $\mu$g m$^{-2}$ yr$^{-1}$) (Larssen et al., 2008) and
Huntington Wildlife forest (15.9 $\mu$g m$^{-2}$ yr$^{-1}$ in deciduous, 26.8 $\mu$g m$^{-2}$ yr$^{-1}$ in conifer)
(Blackwell et al., 2014), the estimated fluxes were higher than in this study.

**4. Conclusions**


Hg in dry and wet deposition, throughfall and litterfall and Hg volatilization from

soil were measured from August 2008 to February 2010 to identify the factors influencing the
amount of atmospheric Hg deposited to forested areas in a temperate deciduous forest in
Korea. In addition measured and theoretical dry deposition were compared. The GOM fluxes
were low in fall and increased towards the spring. PBM fluxes were lowest in fall and peaked
in summer. The estimated and directly measured deposition fluxes were not significantly
correlated likely due to loss of litter samples by wind or wash-off by rainfall and the fact that
accumulation in leaf stomata was not characterized in the direct dry deposition measurement
technique. The average VWM Hg concentration in throughfall was approximately 2.4 times
higher than in precipitation due to wash off of previously deposited Hg from the foliage. Both

15Page **15** / **28**

were higher in winter due to increased concentrations in snow events relative to rain events
likely due to enhanced scavenging of GOM and PBM. TM in litterfall fluxes were highest in
fall when the leaves were dropped and lowest in summer from June to August. Hg emission
fluxes from soil resulted in a cumulative annual volatilization of 6.8 $\mu$g m$^{-2}$ yr$^{-1}$ of GEM.

Based on this all data, the yearly accumulation of Hg in the deciduous forest was
calculated using two input approaches (total input = throughfall + litterfall or wet deposition
+ dry deposition and total output: emission from soil + TM in soil). Using this approach the
accumulation of Hg were 6.8 and 3.9 $\mu$g m$^{-2}$ yr$^{-1}$ respectively. There are several uncertainties
associated with this study as discuss above. The primary ones include that fact that dry
deposition measured with the surrogate surface does not account for accumulation in leaf
stomata yet this technique yielded a larger flux than to litterfall + throughfall – wet deposition.
Litterfall can be lost from the sampler by wind and Hg can be lost from the collected litter
due to washoff from rainfall due to relatively long sampling periods. The differences in the
approaches suggest that approximately half of the GEM stored in the leaf may be released to
back to the atmosphere. DFCs can alter measured fluxes because they cover the soil
potentially blocking some UV light. In addition, several months of measurements were
missed. Finally grab samples for TM in soil may not capture the true variability in the forest.
Additional work should focus on better quantifying dry deposition, TM in soil water,
overflow rate and biogeochemical recycling within the forest canopy and understory.

**Acknowledgments**

This work was supported by the National Research Foundation of Korea (NRF) of Korea
(NRF-2008-0059001 and NRF-2012 R1A1A2042150), Korea Ministry of Environment
(MOE) as "the Environmental Health Action Program" and Brain Korea 21 (BK21) Plus
Project (Center for Healthy Environment Education and Research).

**463**

**464** **References**

**465**

**466** Bishop, K. H., Lee, Y.-H., Munthe, J., and Dambrine, E.: Xylem sap as a pathway for total
**467** mercury and methylmercury transport from soils to tree canopy in the boreal forest,
**468** Biogeochem., 40, 101-113, 1998.
**469** Blackwell, B. D., Driscoll, C. T., Maxwell, J. A., and Holsen, T. M.: Changing climate alters
**470** inputs and pathways of mercury deposition to forested ecosystems, Biogeochem., 119,
**471** 215-228, 2014.
**472** Blackwell, B. D., and Driscoll, C. T.: Deposition of mercury in forests along a montane
**473** elevation gradient, Environ. Sci. Technol., 49, 5363-5370, 2015.
**474** Blanchard, P., Froude, F., Martin, J., Dryfhout-Clark, H., and Woods, J.: Four years of
**475** continuous total gaseous mercury (TGM) measurements at sites in Ontario, Canada,
**476** Atmospheric Environment, 36, 3735-3743, 2002.
**477** Buehler, S., and Hites, R.: The Great Lakes' integrated atmospheric deposition network,
**478** Environ. Sci. Technol., 36, 354A-359A, 2002.
**479** Carpi, A., and Lindberg, S. E.: Sunlight-mediated emission of elemental mercury from soil
**480** amended with municipal sewage sludge, Environ. Sci. Technol., 31, 2085-2091, 1997.
**481** Choi, H.-D., Sharac, T. J., and Holsen, T. M.: Mercury deposition in the Adirondacks: A
**482** comparison between precipitation and throughfall, Atmos. Environ., 42, 1818-1827, 2008.
**483** Choi, H.-D., and Holsen, T. M.: Gaseous mercury emissions from unsterilized and sterilized
**484** soils: the effect of temperature and UV radiation, Environmental Pollution, 157, 1673-1678,
**485** 2009a.
**486** Choi, H.-D., and Holsen, T. M.: Gaseous mercury fluxes from the forest floor of the
**487** Adirondacks, Environ. Pollut., 157, 592-600, 2009b.
**488** Cocking, D., Rohrer, M., Thomas, R., Walker, J., and Ward, D.: Effects of root morphology
**489** and Hg concentration in the soil on uptake by terrestrial vascular plants, Water, Air, Soil
**490** Pollut., 80, 1113-1116, 1995.
**491** Cohen, M., Artz, R., Draxler, R., Miller, P., Poissant, L., Niemi, D., Ratte, D., Deslauriers, M.,
**492** Duval, R., and Laurin, R.: Modeling the atmospheric transport and deposition of mercury
**493** to the Great Lakes, Environ. Res., 95, 247-265, 2004.
**494** Deguchi, A., Hattori, S., and Park, H.-T.: The influence of seasonal changes in canopy
**495** structure on interception loss: application of the revised Gash model, J. Hydrol., 318, 80-
**496** 102, 2006.
**497** Demers, J. D., Driscoll, C. T., Fahey, T. J., and Yavitt, J. B.: Mercury cycling in litter and soil
**498** in different forest types in the Adirondack region, New York, USA, Ecol. Appl., 17, 1341-
**499** 1351, 2007.
**500** Fitzgerald, W. F., Engstrom, D. R., Mason, R. P., and Nater, E. A.: The case for atmospheric
**501** mercury contamination in remote areas, Environ. Sci. Technol., 32, 1-7, 1998.
**502** Fu, X., Feng, X., Dong, Z., Yin, R., Wang, J., Yang, Z., and Zhang, H.: Atmospheric total
**503** gaseous mercury (TGM) concentrations and wet and dry deposition of mercury at a high-
**504** altitude mountain peak in south China, Atmos. Chem. Phys. Discuss, 9, 1-40, 2009.
**505** Fu, X., Feng, X., Dong, Z., Yin, R., Wang, J., Yang, Z., and Zhang, H.: Atmospheric gaseous
**506** elemental mercury (GEM) concentrations and mercury depositions at a high-altitude
**507** mountain peak in south China, Atmos. Chem. Phys, 10, 2425-2437, 2010.

Gabriel, M. C., Williamson, D. G., Zhang, H., Brooks, S., and Lindberg, S.: Diurnal and
seasonal trends in total gaseous mercury flux from three urban ground surfaces, Atmos.
Environ., 40, 4269-4284, 2006.
Gratz, L. E., Keeler, G. J., and Miller, E. K.: Long-term relationships between mercury wet
deposition and meteorology, Atmos. Environ., 43, 6218-6229, 2009.
Gray, J. E., Theodorakos, P. M., Fey, D. L., and Krabbenhoft, D. P.: Mercury concentrations
and distribution in soil, water, mine waste leachates, and air in and around mercury mines
in the Big Bend region, Texas, USA, Environ. geochem. heal., 37, 35-48, 2015.
Graydon, J. A., St. Louis, V. L., Hintelmann, H., Lindberg, S. E., Sandilands, K. A., Rudd, J.
W., Kelly, C. A., Hall, B. D., and Mowat, L. D.: Long-term wet and dry deposition of total
and methyl mercury in the remote boreal ecoregion of Canada, Environ. Sci. Technol., 42,
519  8345-8351, 2008.
Grigal, D., Kolka, R. K., Fleck, J., and Nater, E.: Mercury budget of an upland-peatland
watershed, Biogeochem., 50, 95-109, 2000.
Grigal, D.: Inputs and outputs of mercury from terrestrial watersheds: a review, Environ.
Revie., 10, 1-39, 2002.
Guo, Y., Feng, X., Li, Z., He, T., Yan, H., Meng, B., Zhang, J., and Qiu, G.: Distribution and
wet deposition fluxes of total and methyl mercury in Wujiang River Basin, Guizhou, China,
Atmos. Environ., 42, 7096-7103, 2008.
Hall, B. D., and St. Louis, V. L.: Methylmercury and total mercury in plant litter
decomposing in upland forests and flooded landscapes, Environ. Sci. Technol., 38, 5010-
529  5021, 2004.
Han, Y.-J., Holsen, T. M., Hopke, P. K., and Yi, S.-M.: Comparison between back-trajectory
based modeling and Lagrangian backward dispersion modeling for locating sources of
reactive gaseous mercury, Environ. Sci. Technol., 39, 1715-1723, 2005.
Huang, J., Liu, Y., and Holsen, T. M.: Comparison between knife-edge and frisbee-shaped
surrogate surfaces for making dry deposition measurements: wind tunnel experiments and
computational fluid dynamics (CFD) modeling, Atmospheric environment, 45, 4213-4219,
536  2011.
Huang, J., and Gustin, M. S.: Uncertainties of Gaseous Oxidized Mercury Measurements
Using KCl-Coated Denuders, Cation-Exchange Membranes, and Nylon Membranes:
Humidity Influences, Environ. Sci. Technol., 49, 6102-6108, 2015.
Iverfeldt, Å.: Mercury in forest canopy throughfall water and its relation to atmospheric
deposition, Water Air Soil Pollut., 56, 553-564, 1991.
Jiskra, M., Wiederhold, J. G., Skyllberg, U., Kronberg, R.-M., Hajdas, I., and Kretzschmar,
R.: Mercury deposition and re-emission pathways in boreal forest soils investigated with
Hg isotope signatures, Environ. Sci. Technol., 49, 7188-7196, 2015.
Keim, R. F., Skaugset, A. E., and Weiler, M.: Temporal persistence of spatial patterns in
throughfall, J. Hydrol. , 314, 263-274, 2005.
Kerbrat, M., Pinzer, B., Huthwelker, T., Gäggeler, H., Ammann, M., and Schneebeli, M.:
Measuring the specific surface area of snow with X-ray tomography and gas adsorption:
comparison and implications for surface smoothness, Atmos. Chem. Phys. , 8, 1261-1275,
550  2008.
Kim, P.-R., Han, Y.-J., Holsen, T. M., and Yi, S.-M.: Atmospheric particulate mercury:
Concentrations and size distributions, Atmos. Environ., 61, 94-102, 2012.
Kim, S.-H., Han, Y.-J., Holsen, T. M., and Yi, S.-M.: Characteristics of atmospheric speciated
mercury concentrations (TGM, Hg (II) and Hg (p)) in Seoul, Korea, Atmos. Environ., 43,
555  3267-3274, 2009.

Kolka, R. K., Nater, E., Grigal, D., and Verry, E.: Atmospheric inputs of mercury and organic
carbon into a forested upland/bog watershed, Water Air Soil Pollut., 113, 273-294, 1999.
Lai, S.-o., Holsen, T. M., Hopke, P. K., and Liu, P.: Wet deposition of mercury at a New York
state rural site: Concentrations, fluxes, and source areas, Atmos. Environ., 41, 4337-4348,
560      2007.
Lai, S.-O., Huang, J., Hopke, P. K., and Holsen, T. M.: An evaluation of direct measurement
techniques for mercury dry deposition, Sci. Total Environ., 409, 1320-1327, 2011.
Landis, M. S., and Keeler, G. J.: Critical evaluation of a modified automatic wet-only
precipitation collector for mercury and trace element determinations, Environ. Sci.
Technol., 31, 2610-2615, 1997.
Landis, M. S., and Keeler, G. J.: Atmospheric mercury deposition to Lake Michigan during
the Lake Michigan mass balance study, Environ. Sci. Technol., 36, 4518-4524, 2002.
Larssen, T., de Wit, H. A., Wiker, M., and Halse, K.: Mercury budget of a small forested
boreal catchment in southeast Norway, Sci. Total Environ., 404, 290-296, 2008.
Lin, C.-J., and Pehkonen, S. O.: The chemistry of atmospheric mercury: a review, Atmos.
Environ., 33, 2067-2079, 1999.
Lindberg, S., Turner, R., Meyers, T., Taylor Jr, G., and Schroeder, W.: Atmospheric
concentrations and deposition of Hg to A deciduous forest atwalker branch watershed,
Tennessee, USA, Water Air Soil Pollut., 56, 577-594, 1991.
Lindberg, S., Hanson, P., Meyers, T. a., and Kim, K.-H.: Air/surface exchange of mercury
vapor over forests—the need for a reassessment of continental biogenic emissions, Atmos.
Environ., 32, 895-908, 1998.
Lindberg, S., Bullock, R., Ebinghaus, R., Engstrom, D., Feng, X., Fitzgerald, W., Pirrone, N.,
Prestbo, E., and Seigneur, C.: A synthesis of progress and uncertainties in attributing the
sources of mercury in deposition, J. H. Environ., 36, 19-33, 2007.
Lindqvist, O., Johansson, K., Bringmark, L., Timm, B., Aastrup, M., Andersson, A.,
Hovsenius, G., Håkanson, L., Iverfeldt, Å., and Meili, M.: Mercury in the Swedish
environment—recent research on causes, consequences and corrective methods, Water, Air,
and Soil Pollution, 55, xi-261, 1991.
Lyman, S. N., Gustin, M. S., Prestbo, E. M., and Marsik, F. J.: Estimation of dry deposition of
atmospheric mercury in Nevada by direct and indirect methods, Environ. Sci. Technol., 41,
587      1970-1976, 2007.
Lyman, S. N., Gustin, M. S., and Prestbo, E. M.: A passive sampler for ambient gaseous
oxidized mercury concentrations, Atmospheric Environment, 44, 246-252, 2010.
Lynam, M. M., and Keeler, G. J.: Comparison of methods for particulate phase mercury
analysis: sampling and analysis, Anal. Bioanal. Chem., 374, 1009-1014, 2002.
Ma, M., Wang, D., Sun, R., Shen, Y., and Huang, L.: Gaseous mercury emissions from
subtropical forested and open field soils in a national nature reserve, southwest China,
Atmos. Environ., 64, 116-123, 2013.
Ma, M., Wang, D., Du, H., Sun, T., Zhao, Z., and Wei, S.: Atmospheric mercury deposition
and its contribution of the regional atmospheric transport to mercury pollution at a national
forest nature reserve, southwest China, Environ. Sci. Poll. Res., 22, 20007-20018, 2015.
Miller, E. K., Vanarsdale, A., Keeler, G. J., Chalmers, A., Poissant, L., Kamman, N. C., and
Brulotte, R.: Estimation and mapping of wet and dry mercury deposition across
northeastern North America, Ecotox., 14, 53-70, 2005.
Munthe, J., Hultberg, H., and Iverfeldt, Å.: Mechanisms of deposition of methylmercury and
mercury to coniferous forests, in: Mercury as a Global Pollutant, Springer, 363-371, 1995.
Petersen, G., Iverfeldt, Å., and Munthe, J.: Atmospheric mercury species over central and
Northern Europe. Model calculations and nordic air and precipitation network for 1987 and
1988, Atmos. Environ., 29, 47-67, 1995.
Peterson, C., and Gustin, M.: Mercury in the air, water and biota at the Great Salt Lake (Utah,
USA), Sci. Total Environ., 405, 255-268, 2008.
Price, A., and Carlyle-Moses, D.: Measurement and modelling of growing-season canopy
water fluxes in a mature mixed deciduous forest stand, southern Ontario, Canada, Agric.
For. Meteorol., 119, 69-85, 2003.
Rea, A. W., Lindberg, S. E., and Keeler, G. J.: Dry deposition and foliar leaching of mercury
and selected trace elements in deciduous forest throughfall, Atmos. Environ., 35, 3453-
613     3462, 2001.
Risch, M. R., DeWild, J. F., Krabbenhoft, D. P., Kolka, R. K., and Zhang, L.: Litterfall
mercury dry deposition in the eastern USA, Environmental Pollution, 161, 284-290, 2012.
Rolfhus, K., Sakamoto, H., Cleckner, L., Stoor, R., Babiarz, C., Back, R., Manolopoulos, H.,
and Hurley, J.: Distribution and fluxes of total and methylmercury in Lake Superior,
Environ. Sci. Technol., 37, 865-872, 2003.
Schroeder, W. H., and Munthe, J.: Atmospheric mercury—an overview, Atmos. Environ., 32,
620     809-822, 1998.
Schwesig, D., and Matzner, E.: Pools and fluxes of mercury and methylmercury in two
forested catchments in Germany, Sci. Total Environ., 260, 213-223, 2000.
Selin, N. E., Jacob, D. J., Park, R. J., Yantosca, R. M., Strode, S., Jaeglé, L., and Jaffe, D.:
Chemical cycling and deposition of atmospheric mercury: Global constraints from
observations, Journal of Geophysical Research: Atmospheres, 112, 2007.
Selvendiran, P., Driscoll, C. T., Montesdeoca, M. R., and Bushey, J. T.: Inputs, storage, and
transport of total and methyl mercury in two temperate forest wetlands, J. Geophys. Res.,
628     113, 2008.
Seo, Y.-S., Han, Y.-J., Choi, H.-D., Holsen, T. M., and Yi, S.-M.: Characteristics of total
mercury (TM) wet deposition: scavenging of atmospheric mercury species, Atmos.
Environ., 49, 69-76, 2012.
Seo, Y.-S., Han, Y.-J., Holsen, T. M., Choi, E., Zoh, K.-D., and Yi, S.-M.: Source
identification of total mercury (TM) wet deposition using a Lagrangian particle dispersion
model (LPDM), Atmos. Environ., 104, 102-111, 2015.
Shanley, J. B., Engle, M. A., Scholl, M., Krabbenhoft, D. P., Brunette, R., Olson, M. L., and
Conroy, M. E.: High mercury wet deposition at a "clean air" site in Puerto Rico, Environ.
Sci. Technol., 49, 12474-12482, 2015.
Sigler, J., Mao, H., and Talbot, R.: Gaseous elemental and reactive mercury in Southern New
Hampshire, Atmospheric Chemistry and Physics, 9, 1929-1942, 2009.
Skinner, D.: UV curing through semi-transparent materials: the challenge of the DVD
bonding process, RADTECH-NORTH AMERICA-, 1998, 140-146.
St. Louis, V. L., Rudd, J. W., Kelly, C. A., Hall, B. D., Rolfhus, K. R., Scott, K. J., Lindberg,
S. E., and Dong, W.: Importance of the forest canopy to fluxes of methyl mercury and total
mercury to boreal ecosystems, Environ. Sci. Technol., 35, 3089-3098, 2001.
U.S.EPA: Persistent, bioaccumulative and toxic chemical program. http://
www.epa.gov/pbt, 1997a.
U.S.EPA: U.S. EPA Lake Michigan Mass Balance Methods Compendium.
http://nepis.epa.gov/, 1997b.
U.S.EPA: Method 1631, Revision E: Mercury in Water by Oxidation, Purge and Trap, and
Cold Vapor Atomic Fluorescence Spectrometry, 2002.
Wallschläger, D., Herbert Kock, H., Schroeder, W. H., Lindberg, S. E., Ebinghaus, R., and
Wilken, R.-D.: Mechanism and significance of mercury volatilization from contaminated
floodplains of the German river Elbe, Atmos. Environ., 34, 3745-3755, 2000.
Wang, S., Feng, X., Qiu, G., Wei, Z., and Xiao, T.: Mercury emission to atmosphere from
Lanmuchang Hg–Tl mining area, southwestern Guizhou, China, Atmos. Environ., 39,
656      7459-7473, 2005.
Wang, Y., Peng, Y., Wang, D., and Zhang, C.: Wet deposition fluxes of total mercury and
methylmercury in core urban areas, Chongqing, China, Atmos. Environ., 92, 87-96, 2014.
Weiss-Penzias, P. S., Gay, D. A., Brigham, M. E., Parsons, M. T., Gustin, M. S., and ter
Schure, A.: Trends in mercury wet deposition and mercury air concentrations across the US
and Canada, Sci. Total Environ., 2016.
Zhang, L., Blanchard, P., Gay, D., Prestbo, E., Risch, M., Johnson, D., Narayan, J., Zsolway,
R., Holsen, T., and Miller, E.: Estimation of speciated and total mercury dry deposition at
monitoring locations in eastern and central North America, Atmos. Chem. Phys. , 12, 4327-
665      4340, 2012.
Zhu, J., Wang, T., Talbot, R., Mao, H., Yang, X., Fu, C., Sun, J., Zhuang, B., Li, S., and Han,
Y.: Characteristics of atmospheric mercury deposition and size-fractionated particulate
mercury in urban Nanjing, China, Atmos. Chem. Phys. , 14, 2233-2244, 2014.


**Table List**
Table 1. Cumulative precipitation depths, VWM Hg concentration, cumulative Hg fluxes in
precipitation and throughfall during leaf-on and leaf-off periods.



**Figure List**
Fig. 1. The locations of the sampling sites used in this study (Yangsu-ri, Korea)
Fig. 2. Seasonal variation in dry deposition flux for GOM and PBM under the deciduous
forest.
Fig. 3. Seasonal variation in VWM TM concentration, rainfall depth and TM flux in
precipitation and throughfall.
Fig. 4. Seasonal variation in TM concentration and flux in a deciduous forest.
Fig. 5. The estimated annual Hg emission fluxes in 2009 from soil.
Fig. 6. Comparison of deposition flux calculated by summing wet deposition + dry deposition
and throughfall + litterfall


Table 1. Cumulative precipitation depths, VWM Hg concentration, cumulative Hg fluxes in
precipitation and throughfall during leaf-on and leaf-off periods.

| | Cumulative precipitation depth (mm) | | VWM Hg Concentration (ng L$^{-1}$) | | Cumulative Hg fluxes ($\mu g$ Hg m$^{-2}$) | |
| --- | --- | --- | --- | --- | --- | --- |
| | Leaf-on | Leaf-off | Leaf-on | Leaf-off | Leaf-on | Leaf-off |
| Precipitation | 968.3 | 117.6 | 5.4 | 7.2 | 3.8 | 0.5 |
| Throughfall | 1009.7 | 114.7 | 8.1 | 18.3 | 4.9 | 1.8 |




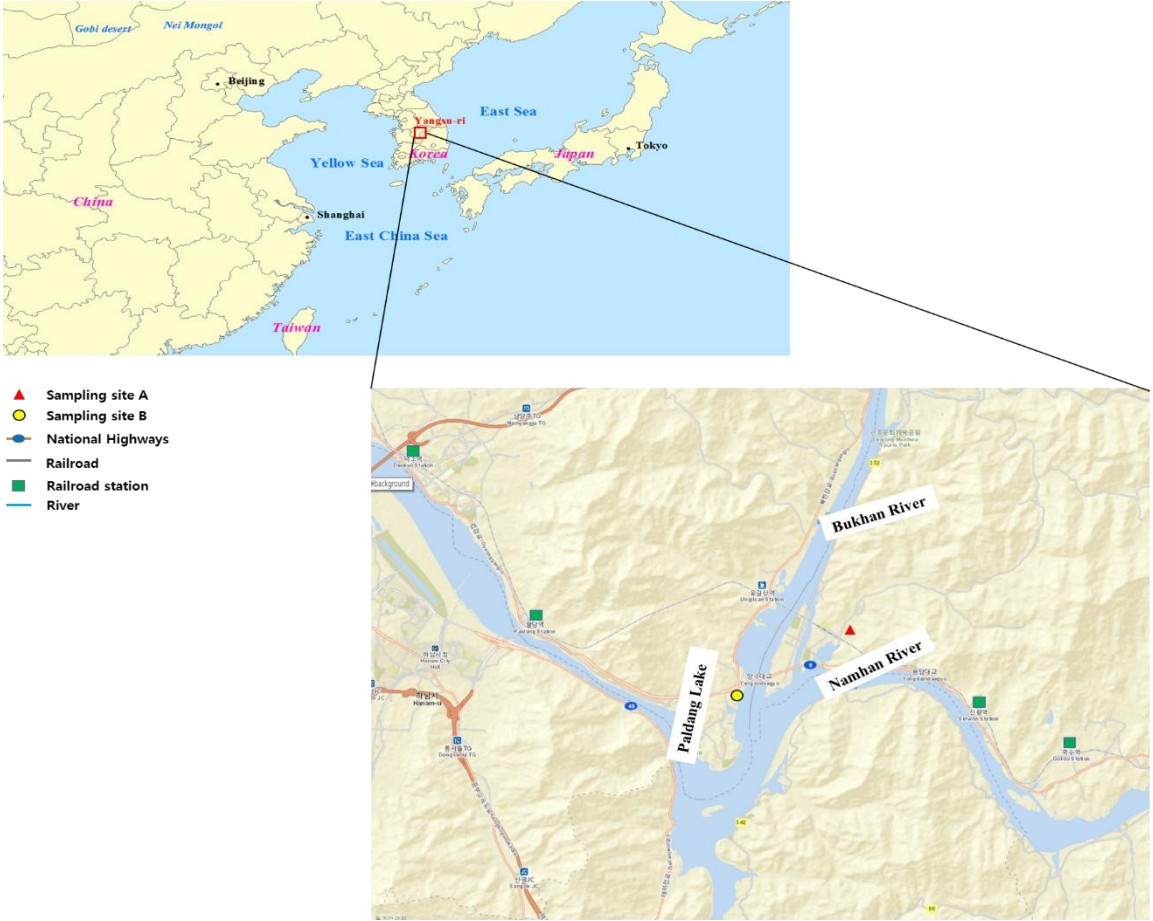


**Fig. 1. The locations of the sampling sites used in this study (Yangsu-ri, Korea).**



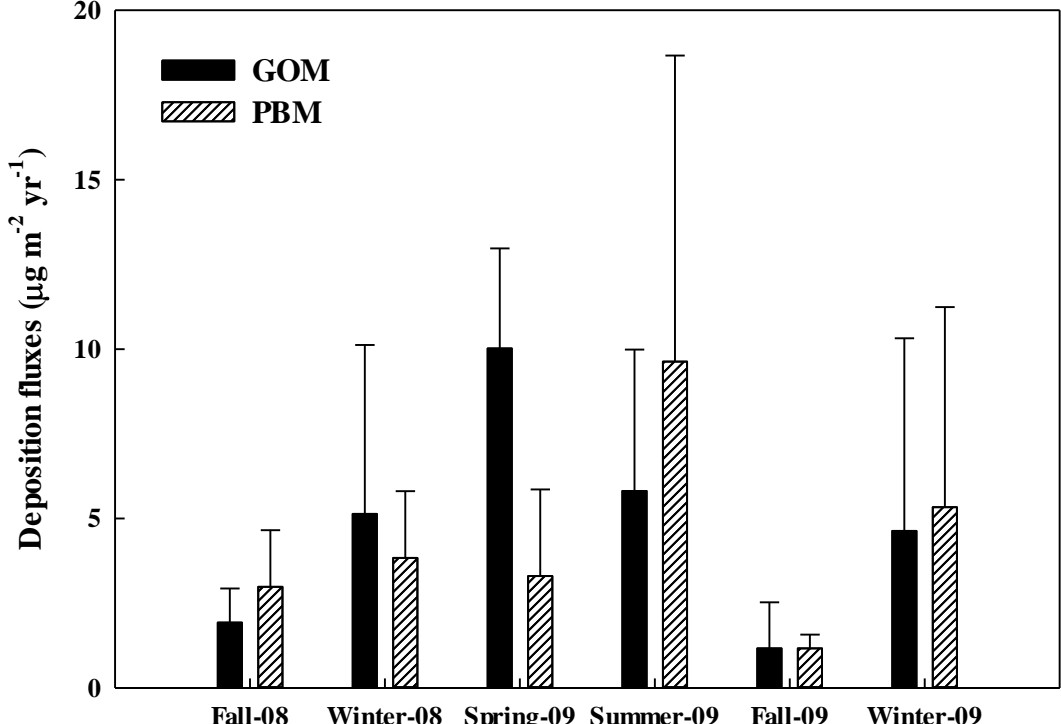


**Fig. 2. Seasonal variation in dry deposition flux for GOM and PBM under the**
**deciduous forest.**


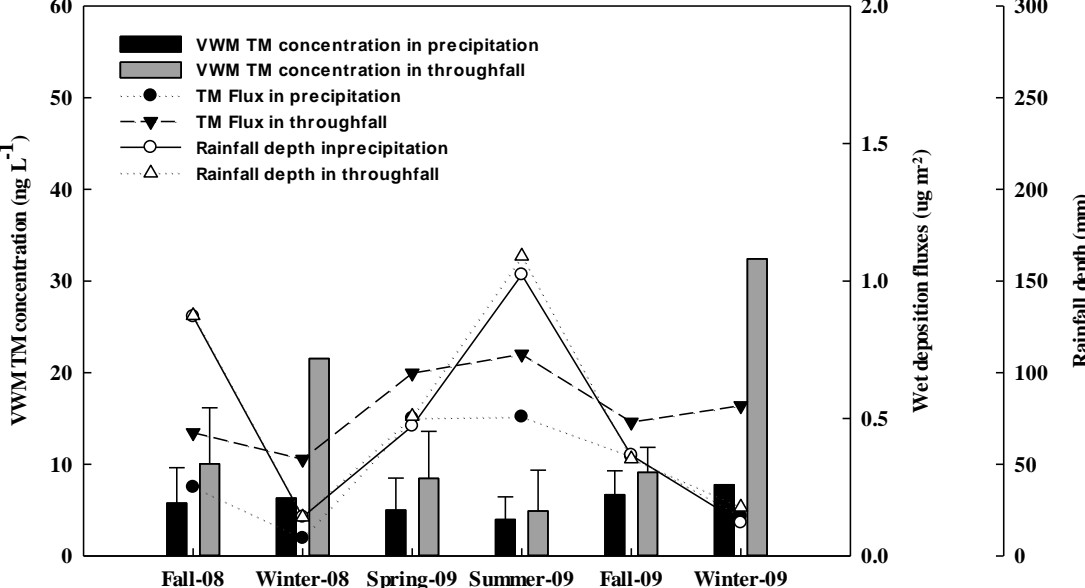


**Fig. 3. Seasonal variation in VWM TM concentration, rainfall depth and TM flux in**
**precipitation and throughfall.**



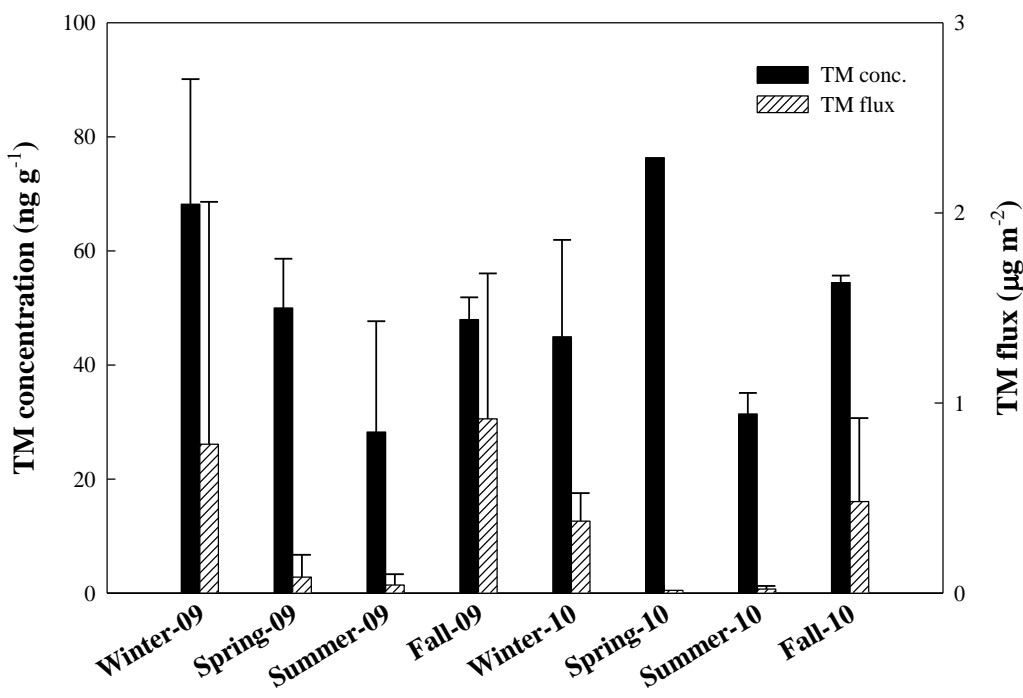


**Fig. 4. Seasonal variation in TM concentration and flux in a deciduous.**





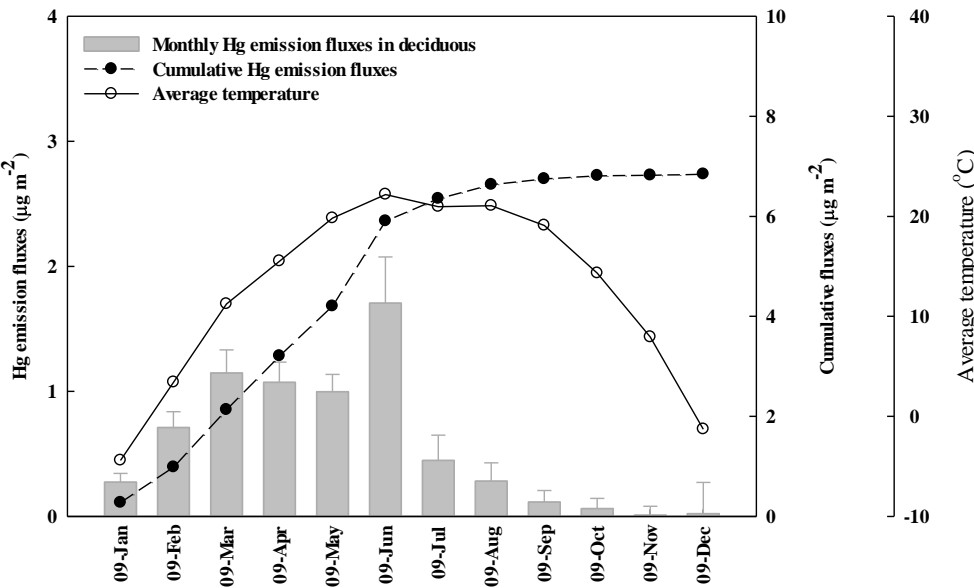


**Fig. 5. The estimated annual Hg emission fluxes in 2009 from soil.**


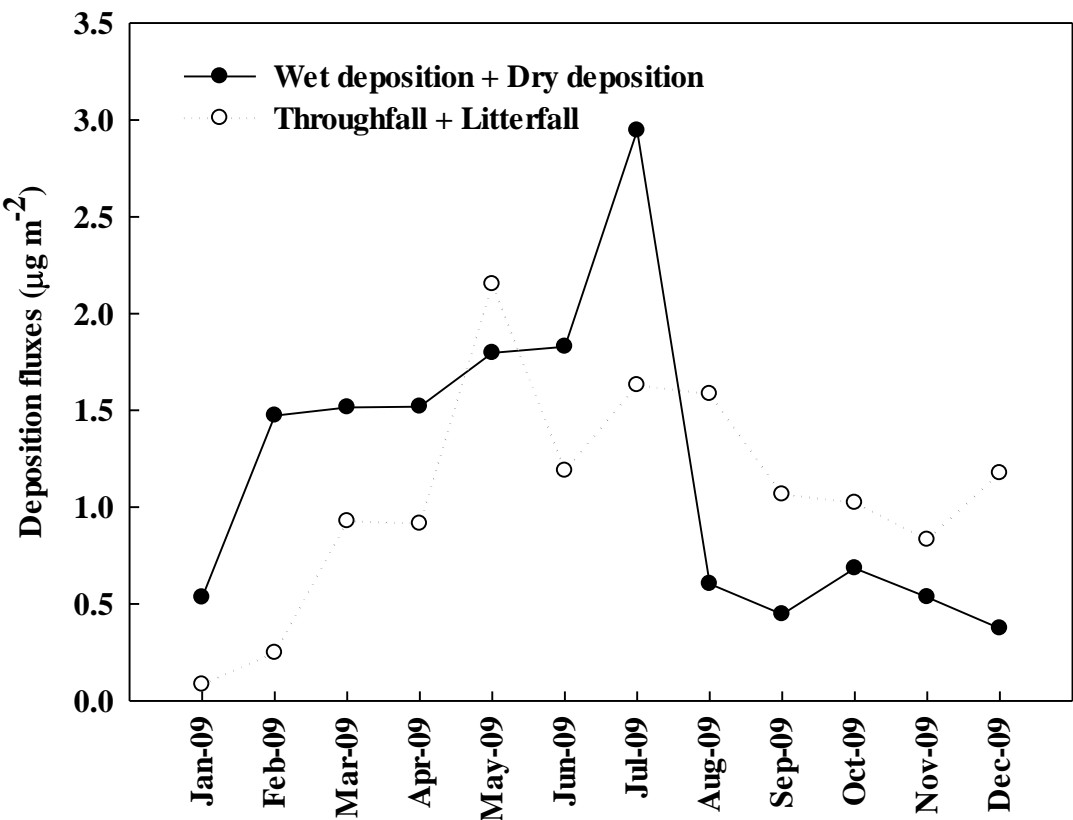


**Fig. 6. Comparison of deposition flux calculated by summing wet deposition + dry**
**deposition and throughfall + litterfall**