# Peer review of "Total Atmospheric Mercury Deposition in Forested Areas in Korea Jin-Su Han1, Yong-Seok Seo1, 2, Moon-Kyung Kim1, 2, Thomas M. Holsen3, Seung-Muk Yi1, 2, \* 1 Department of Environmental Health Sciences, Graduate School of Public H"

_Atmospheric Chemistry and Physics, 2016_

## Referee Comment (RC1) · Anonymous Referee #1 · 8 Feb 2016

07 February 2016

**Review of Jin-Su Han et al. "Total Atmospheric Mercury Deposition in Forest Areas in Korea"**

**General Comments**
Han and co-authors present a site-specific study of mercury deposition to a forested location in Korea. Based on measured fluxes, the authors construct an estimated mass budget for the study site. Measurements in this region of the world are sparse and additional datasets are a welcome contribution to the mercury literature. The analysis in the manuscript, however, needs to be matured. It presently reads like a core dump of numbers and the analysis feels lacking. I recommend the manuscript be revised before publication. The authors should focus on providing more insight to the readers.

**Specific Comments**

**Abstract:** It's just a list of numbers. What's missing is why the authors did the study and why the results they found matter. The Abstract needs a punch line.

**Introduction:** Could be greatly improved by including a clearer statement of the problem or scientific question they're trying to answer with this dataset. The logical progression of the Introduction is a little hard to follow and it doesn't build a clear storyline for the rest of the paper.

**Methods, Site Description:** The authors need a clear statement of why this particular location in Korea was selected. Lines 117-120 provided somewhat of an explanation, but it feels too vague. What does this site tell us that other sites don't?

**Section 2.4 QA/QC:** Too many acronyms are introduced. Makes the text difficult to follow.

**Results and Discussion:** A few recommendations here:
- The authors use a knife-edge surrogate surface for PBM and GOM dry deposition measurements (Section 2.2.1). It would be useful to provide some discussion on how this method compares to other surrogate surface methods (e.g., the work done by Mae Gustin's).
- **Page 8, lines 220-228:** If all of these numbers are important, I suggest condensing into a table. It's difficult to parse text right now.
- **Page 6, lines 240-243:** The importance of this paragraph is unclear. Could it be deleted?
- **Page 9, lines 260-270:** Which explanation do the authors think is most plausible? The text currently gives the impression the authors are just guessing. A more thoughtful scrutiny of the proposed explanations would be welcome.
- **Page 10, line 282:** "Therefore, all of the Hg deposited…" What fraction is lost? What fraction is retained? This could be really interesting.
- **Page 10, lines 294-296:** "… the rest of the variation is likely due to variations in local…" The "rest" here being >80%, correct? A more rigorous explanation of the majority of the

variability seen in the data would be helpful. Being able to explain less than 20% does not give confidence in the interpretation.

- **Page 10, lines 307-309:** Why is an $r^2=0.59$ ($p<0.05$) "significantly correlated" and $r^2=0.56$ ($p<0.05$) "moderately correlated"?
- **Page 12:** If you are missing data in January and February, the stated assumption that "fluxes were assumed to be equal to the average of the flux of the month before" doesn't make sense. How are you handling consecutive months of missing data?
- **Page 13, lines 382-383:** It would be useful here to be specific and describe what "both input approaches" are. It's not that clear what approaches you mean.
- **Section 3.8:** This section is disjointed and lacks cohesion. Revision strongly encouraged, with a focus on building a logical progression.

**Conclusions:** The manuscript needs a Conclusions section. Without Conclusions, the manuscript incomplete and doesn't seem mature enough for publication. A couple of strong synthesis statements from the authors about why their results add to our knowledge in the mercury field would really help the paper.

**Supporting Information:** I encourage the authors to make their data available in the SI. This will make it easier for other interested scientists, especially modelers, to compare against the data in Korea. The mercury community will be excited about this dataset and want to weave it into their comparisons – make it easy for them!

---

## Referee Comment (RC2) · Anonymous Referee #2 · 11 Feb 2016

Review of acp-2016-7, Total Atmospheric Mercury Deposition in Forest Areas in Korea By Han et al., 2016 ACPD General comments: Han et al. report mercury (Hg) deposition to forest in Korea. I read this study like a routine measurement with discontinuous time periods. The way this manuscript was written more like a data report without detail discussions and data analysis. Most references are also out of date. There is no cutting edge research in current manuscript. I suggest the authors re-consider the manuscript structure and investigate the data in more detail way than current data analysis.

Specific comments: Abstract is written in the way with number reporting, there is no significant conclusions and any new discovery. After reading the abstract, I am not attracted by this abstract. Most of these measurements have been done somewhere else on the earth, what is the significant finding here? What will the data help global or regional mercury research? If the authors just want to submit a routine report, they

should consider some regional environmental journals. Line 69-71, the original papers were not cited Line 72-74, re-write, PBM is particle-bound Hg, how can it be adsorbed on PM? You could say oxidized mercury or GOM. Line 77-79, add Selin et al., 2007 and Lindberg et al., 2007 Line 84-86, not clear, also update the reference here Line 88-89, please be more clear, how does uptake via roots impact Hg deposition. Also stomatal uptake of Hg0 emitted from soils? I don't understand this sentence. Line 133, please discuss problems from using KCl coated quartz surface. Lyman et al., 2010; Huang et al., 2013/2015, McClure et al., 2015, Lynam and Keeler 2006 Sampling method, what are the time periods? Analytical method, did the author develop the thermal desorption method? If not please cite references. If I understand this correctly, KCl QFF was heated to 525C and QFF was heated to 900C to separate GOM and PBM. Two questions here. 1. Is dry deposition collected up facing or down facing? and how up/down facing impact measurement? 2. Is this possible for GOM attach on QFF and quantified as PBM, and PBM attach on KCl-QFF and quantified as GOM? What is the recovery for the thermal desorption system? Recovery for Tekran 2537 direct injection 87% is too low usually from 93-107%. How many sampling time periods? Only 4 field blanks? Why? Volatilization from soil, what are MDL or blanks? Section 3.1, if you only have a short time period during each season, how can you really see the seasonal pattern? Please add more detail information for sampling plan. What statistical test are you using, please add information for every place you mention significant difference. Line 281, what is mechanical weathering?

---

## Author Comment (AC1) · 4 May 2016

[Comment 1] Abstract: It's just a list of numbers. What's missing is why the authors did the study and why the results they found matter. The Abstract needs a punch line. -> we revised the Abstract as follows on Line 34 to Line 51

[Comment 2] Introduction: Could be greatly improved by including a clearer statement of the problem or cientific question they're trying to answer with this dataset. The logical progression of the Introduction is a little hard to follow and it doesn't build a clear storyline for the rest of the paper. ->The introduction has been modified as follows Line 58 to Line 114.

[Comment 3] Methods, Site Description: The authors need a clear statement of why this particular location in Korea was selected. Lines 117-120 provided somewhat of an

[Figure]

explanation, but it feels too vague. what does this site tell us that other sites don't? -> We added the following information about the sampling site as follows on Line 127 to Line 133.

[Comment 4] Section 2.4. QA/QC: Too many acronyms are introduced. Makes the text difficult to follow. ->We revised section 2.4. QA/QC as follows on Line 209 to Line 221.

[Comment 5] The authors use a knife-edge surrogate surface for PBM and GOM dry deposition measurements (Section 2.2.1). It would be useful to provide some discussion on how this method compares to other surrogate surface methods (e.g., the work done by Mae Gustin's). -> We provied some additional discussion and refer to Gustin et al. (2016) as follows on Line 141 to 154. Additional changes were made to this section based on Reviewer 2 comments.

[Comment 6] Page 8, lines 220-228: If all of these numbers are important, I suggest condensing into a table. It's difficult to parse text right now. -> We revised seasonal dry deposition data as follows on Line 241 to Line 247.

[Comment 7] Page 8, lines 240-243: The importance of this paragraph is unclear. Could it be deleted? -> W deleted Page 8, lines 240-243

[Comment 8] Page 9, lines 260-270: Which explanation do the authors think is most plausible? The text currently gives the impression the authors are just guessing. A more thoughtful scrutiny of the proposed explanations would be welcome. ->This section has been revised as follows on Line 276 to Line 285.

[Comment 9] Page 10, line 282: "Therefore, all of the Hg deposited..." What fraction is lost? What fraction is retained? This could be really interesting. -> Unfortunately we did not make any direct measurements of what was collected on the leaves and how much remained after a precipiation event so we can not address this question. However we did add a bit more discussion on Line 296 to Line 300.

[Comment 10] Page 10, lines 294-296: "... the rest of the variation is likely due to

variations in local. . ." The "rest" here being >80%, correct? A more rigorous explanation of the majority of the variability seen in the data would be helpful. Being able to explain less than 20% does not give confidence in the interpretation. ->We added further explanations as follows on Line 312 to Line 321.

[Comment 11] Page 10, lines 307-309: Why is an r2=0.59 (p<0.05) "significantly correlated" and r2=0.56 (p<0.05) "moderately correlated"? -> Those should be the same - we corrected "moderately correlated" to "significantly correlated" as follows on Line 330.

[Comment 12] Page 12: If you are missing data in January and February, the stated assumption that "fluxes were assumed to be equal to the average of the flux of the month before" doesn't make sense. How are you handling consecutive months of missing data? -> We corrected this mistake about missing data as follows on Line 377 to Line 380.

[Comment 13] Page 13, lines 382-383: It would be useful here to be specific and describe what "both input approaches" are. It's not that clear what approaches you mean. -> We added further explanations as follows on Line 406 to Line 408 : "The yearly estimated budget of Hg was calculated using both input approaches (Total input = wet deposition + dry deposition or Total input = throughfall + litterfall) as follows..."

[Comment 14] Section 3.8: This section is disjointed and lacks cohesion. Revision strongly encouraged, with a focus on building a logical progression. -> This section has been revised as follows on Line 406 to Line 419.

[Comment 15] Conclusions: The manuscript needs a Conclusions section. Without Conclusions, the manuscript incomplete and doesn't seem mature enough for publication. A couple of strong synthesis statements from the authors about why their results add to our knowledge in the Hg field would really help the paper. -> We added a conclusions section as follows on Line 423 to Line 451.

[Comment 16] Supporting Information: I encourage the authors to make their data available in the SI. This will make it easier for other interested scientists, especially modelers, to compare against the data in Korea. The mercury community will be excited about this dataset and want to weave it into their comparisons – make it easy for them! -> We revised SI as follows on Line 31 to Line 50.

Please also note the supplement to this comment:
http://www.atmos-chem-phys-discuss.net/acp-2016-7/acp-2016-7-AC1-supplement.zip

---

## Author Comment (AC2) · 4 May 2016

[Comment 1] Abstract is written in the way with number reporting, there is no significant conclusions and any new discovery. -> We revised the Abstract as follows on Line 34 to Line 51.

[Comment 2] Line 69-71, the original papers were not cited and PBM is particle-bound Hg, how can it be adsorbed on PM? You could say oxidized Hg or GOM. -> We revised text and reference as follows on Line 70 to Line 71 : "PBM is created by GEM or GOM adsorbing to a particle (Lai et al., 2011)."

[Comment 3] Line 72-74, re-write. -> We revised as follows on Line 71 to Line 74 : "...... Atmospheric PBM transport is significantly affected by its particle size distribution and may contribute to both wet and dry deposition (Lynam and Keeler, 2002). Wet and

dry deposition of atmospheric Hg is an important input to the aquatic and terrestrial ecosystems"

[Comment 4] Line 84-86, not clear, also update the reference here Line 88-89 -> Wwe revised and updated this text as follows on Line 80 to Line 105.

[Comment 5] Line 77-79, add Selin et al., 2007 and Lindberg et al., 2007 -> We added Selin et al., 2007 and Lindberg et al., 2007 as follows on Line 77 to Line 79 : "...., resulting in adverse health and ecological effects (Ma et al., 2013; Lindberg et al., 2007; Rolfhus et al., 2003; Selin et al., 2007; Weiss-Penzias et al., 2016; Zhu et al., 2014)."

[Comment 6] how does uptake via roots impact Hg deposition. Also stomatal uptake of Hg0 emitted from soils? -> We added information as follows on Line 102 to Line 105 : " .... root uptake of dissolved Hg in soil and soil water and stomatal uptake of GEM that was volatilized from soils (Bishop et al., 1998; Cocking et al., 1995; Ma et al., 2015; St. Louis et al., 2001). Recycled Hg would increase throughfall and litterfall concentrations (St Louis. et al., 2001)."

[Comment 7] I don't understand this sentence. Line 133, please discuss problems from using KCl coated quartz surface. Lyman et al., 2010; Huang et al., 2013/2015, McClure et al., 2015, Lynam and Keeler 2006 Sampling method, what are the time periods? -> We revised and discussed problems from using KCl coated quartz surface as follows on Line 141 to Line 154.

[Comment 8] Analytical method, did the author develop the thermal desorption method? If not please cite references. -> We added references as follows on Line 187 to Line 188 : "....... zero air passed through until the Hg concentration was zero (Kim et al., 2009; Kim et al., 2012)."

[Comment 9] If I understand this correctly, KCl QFF was heated to 525C and QFF was heated to 900C to separate GOM and PBM. Two questions here. 1. Is dry deposition

collected up facing or down facing? and how up/down facing impact measurement? 2. Is this possible for GOM attach on QFF and quantified as PBM, and PBM attach on KCl-QFF and quantified as GOM? -> We added information about sampling method of dry deposition as follows on Line 141 to Line 154 and Line 192 to Line 193.

[Comment 10] What is the recovery for the thermal desorption system? Recovery for Tekran 2537 direct injection 87% is too low usually from 93-107%. How many sampling time periods? Only 4 field blanks? Why? Volatilization from soil, what are MDL or blanks? -> We revised section 2.4.1 and 2.4.4 as follows on Line 209 to Line 214 and Line 232 to Line 235.

[Comment 11] Section 3.1, if you only have a short time period during each season, how can you really see the seasonal pattern? Please add more detail information for sampling plan. -> We added information about sampling plan as follows on Line 241 to Line 242 : "Weekly sample were collected using quartz (PBM) and KCl coated quartz filters (GOM)"

[Comment 12] What statistical test are you using, please add information for every place you mention significant difference. -> We added information about significant difference as follows on Line 247, Line 265, Line 272, Line 290, Line 327, Line 396.

[Comment 13] Line 281, what is mechanical weathering? -> We revised as follows on Line 294 to Line 296 : "Other possible sources of Hg in throughfall are leaching and biogeochemical recycling of Hg from foliage (St. Louis et al., 2001)."

[Comment 14] Most references are also out of date. -> We added recent references.

Please also note the supplement to this comment:
http://www.atmos-chem-phys-discuss.net/acp-2016-7/acp-2016-7-AC2-supplement.zip

---

## Author Response (AR1)

May 5, 2016

Dear Editor,

We appreciate the reviewers' suggestions which have considerably improved the manuscript (**acp-2016-7**). Enclosed are point-by-point responses to the reviewers. We hope that with these changes the manuscript will be suitable for publication in "**Atmospheric Chemistry**

**and Physics**"

Thank you very much.

Sincerely,

Seung-Muk Yi

Professor, Dept. of Environmental Health, Graduate School of Public Health

Seoul National University, 1 Gwanak-ro, Gwanak-gu, Seoul 151-742, South Korea

Telephone: (82) 2-880-2736, Fax: (82) 2-762-9105, E-mail: yiseung@snu.ac.kr

**Response to Reviewers' Comments**

● Journal: ACP

● Title: Total Atmospheric Mercury Deposition in Forested Areas in Korea

● Author(s): Jin-Su Han, Yong-Seok Seo, Moon-Kyung Kim, Thomas M. Holsen, Seung-Muk Yi

● MS No.: acp-2016-7

● MS Type: Research article

● Special Issue: Data collection, analysis and application of speciated atmospheric mercury

**Response to Referee 1:**

**Comment 1**

**Abstract:** It's just a list of numbers. What's missing is why the authors did the study and why the results they found matter. The Abstract needs a punch line.

    **Response 1**

As suggested, we revised the Abstract as follows on **Line 34 to Line 51**

*"In this study, mercury (Hg) in dry and wet deposition, throughfall and litterfall, and Hg volatilization from soil were measured from August 2008 to February 2010 to identify the factors influencing the amount of atmospheric Hg deposited to forested areas in a temperate deciduous forest in Korea. For this location there was no significant correlation between the estimated dry deposition flux (litterfall + throughfall – wet deposition) (6.7 $\mu g\ m^{-2}\ yr^{-1}$) and directly measured dry deposition (9.9 $\mu g\ m^{-2}\ yr^{-1}$) likely due primarily to Hg losses from the litterfall collector. Dry deposition fluxes in cold seasons (fall and winter) were lower than in warmer seasons (spring and summer). The volume-weighted mean (VWM) Hg concentrations in both precipitation and throughfall were highest in winter likely due to increased scavenging by snow events. Since Korea experiences abundant rainfall in summer, VWM Hg concentrations in summer were lower than in other seasons. Litterfall fluxes were highest in the late fall to early winter when leaves were dropped from the trees (September to November). The cumulative annual Hg emission flux from soil was 6.8 $\mu g\ m^{-2}\ yr^{-1}$. Based on these data, the yearly deposition fluxes of Hg calculated using two input approaches (throughfall + litterfall or wet deposition + dry deposition), were 6.8 and 3.9 $\mu g\ m^{-2}\ yr^{-1}$ respectively. This is the first reported study which measured the amount of atmospheric Hg deposited to forested areas in Korea and thus our results provide useful information related to Hg fate and transport in this part of the world."*

**Comment 2.**

**Introduction:** Could be greatly improved by including a clearer statement of the problem or scientific question they're trying to answer with this dataset. The logical progression of the Introduction is a little hard to follow and it doesn't build a clear storyline for the rest of the paper.

As suggested, the introduction has been modified as follows **Line 58 to Line 114**.

[revised manuscript text omitted]

**Comment 3.**

**Methods, Site Description:** The authors need a clear statement of why this particular location in Korea was selected. Lines 117-120 provided somewhat of an explanation, but it feels too vague. what does this site tell us that other sites don't?

**Response 3**

As suggested, we added the following information about the sampling site as follows on **Line**

**127 to Line 133**.

*"Dry deposition, throughfall, litterfall, volatilization from soils and TM in soil samples were*

*determined in a deciduous forest including primarily chestnut (Elevation 60 m, N37°32´,*

*E127°20´) (site B in Fig. 1) about 2 km away from site A. This area contains rivers, a flood*

*plain, agricultural land, residential areas, forests, and wetlands that are expected to actively*

*methylate Hg. Therefore, the study sites are appropriate for identifying the in/out flow of Hg*

*in a forested ecosystem typical for this part of the world."*

**Comment 4.**

**Section 2.4. QA/QC:** Too many acronyms are introduced. Makes the text difficult to follow.

**Response 4.**

As suggested, we revised section 2.4. QA/QC as follows on **Line 209 to Line 221**.

*"A automated daily calibration of Tekran 2537 routinely operated was performed using an*

*internal permeation source. Two-point calibrations (zero and span) were operated separately*

*for each pure gold cartridge. A recovery of $102 \pm 2.9\%$ ($r^2 > 0.9995$) ($n = 4$) measured by*

*directly injecting knowing amounts of five Hg standards which was connected to zero air.*

*The Method Detection Limit (MDL) was by measuring the Hg concentration in zero air was*

*$0.04$ ng m$^{-3}$. More additional information is provied SI."*

*"Quality assurance and quality control were based on the U.S. EPA Methods 1631 version E*

*(U.S.EPA, 2002) and LMMBMC (U.S.EPA., 1994a). The MDL (three times the standard*

*deviation of seven sequential reagent blanks) for TM in wet deposition and throughfall was*

*$0.05$ ng L$^{-1}$. The standard curve was acceptable when r2 was greater than 0.9995 (linear).*

*More additional information is described SI."*

**Comment 5.**

The authors use a knife-edge surrogate surface for PBM and GOM dry deposition measurements (Section 2.2.1). It would be useful to provide some discussion on how this method compares to other surrogate surface methods (e.g., the work done by Mae Gustin's).

**Response 5.**

As suggested, we provied some additional discussion and refer to Gustin et al. (2016) as follows **on Line 141 to 154**. Additional changes were made to this section based on Reviewer 2 comments.

*"Some studies have investigated the use of surrogate surfaces to directly measure Hg dry deposition (Lyman et al., 2007; Peterson and Gustin, 2008). Surrogate surfaces allow better control over exposure times than those provided with natural vegetation (Lai et al., 2011). Surrogate surfaces with cation exchange membranes have been useful for measuring GOM however they may collect a very small aerosol fraction by diffusion (Lyman et al., 2007; Huang and Gustin, 2015b). Similar to previous studies (Lai et al., 2011; Yi et al., 1996), in this project the dry deposition sampler was equipped with a knife-edge surrogate surface (KSS) sampler with the collection media facing up. Forty seven-mm quartz filters were used to measure PBM deposition and KCl-coated quartz filters were used to measure GOM + PBM deposition. The quartz filter and KCl-coated quartz filter (soaked in KCl solution for 12h and dried on clean bench) were pre-baked in a quartz container at 900 ºC for PBM and 525 ºC for GOM + PBM. Before weekly sampling, the filters were placed on a filter holder base and held in place with a retaining ring and then was deployed in the KSS. Filter exposed to the atmosphere from approximately one week and two side-by-side samples were deployed during each dry day."*

**Comment 6.**

**Page 8, lines 220-228:** If all of these numbers are important, I suggest condensing into a table. It's difficult to parse text right now.

**Response 6.**

As suggested, we revised seasonal dry deposition data as follows on **Line 241 to Line 247**.

*"Weekly samples were collected using quartz filters (PBM) and KCl coated quartz filters*

*(GOM). The average dry deposition fluxes for GOM (Table S1) and PBM (Table S2) were 5.4*

*$\mu g$ $m^{-2}$ $yr^{-1}$ (range: 0.4 ~ 14.4 $\mu g$ $m^{-2}$ $yr^{-1}$) and 4.3 $\mu g$ $m^{-2}$ $yr^{-1}$ (range: 0.8 ~ 19.4 $\mu g$ $m^{-2}$ $yr^{-1}$),*

*respectively. The dry deposition fluxes for GOM were highest in spring 2009 (10.0 $\pm$ 2.0 $\mu g$*

*$m^{-2}$ $yr^{-1}$), lowest in fall 2009 (1.2 $\pm$ 1.4 $\mu g$ $m^{-2}$ $yr^{-1}$) while the dry deposition fluxes for PBM*

*were highest in summer 2009 (9.6 $\pm$ 9.0 $\mu g$ $m^{-2}$ $yr^{-1}$), lowest in fall 2009 (1.2 $\pm$ 0.4 $\mu g$ $m^{-2}$ $yr^{-1}$)*

*(Fig. 2)."*

**Comment 7.**

**Page 8, lines 240-243:** The importance of this paragraph is unclear. Could it be deleted?

  **Response 7.**

As suggested we deleted Page 8, lines 240-243

**Comment 8.**

**Page 9, lines 260-270:** Which explanation do the authors think is most plausible? The text currently gives the impression the authors are just guessing. A more thoughtful scrutiny of the proposed explanations would be welcome.

  **Response 8.**

As suggested, this section has been revised as follows on **Line 276 to Line 285**.

*"The high VWM Hg concentrations in precipitation and throughfall in winter were*

*associated with the combined effects of reduced mixing heights which increases atmospheric*

*concentrations (Kim et al., 2009; Seo et al., 2015), low rainfall depth (11.7% of total rainfall*

*depth) which is a typical pattern in Yangpyung, Korea (KMA,*

*http://www.kma.go.kr/weather/climate/average_30years.jsp?yy_st&tnqh_x003D;2011&*

*stn&tnqh_x003D;108&norm&tnqh_x003D;M&obs&tnqh_x003D;0&mm&tn*

*qh_x003D;5&dd&tnqh_x003D;25&x&tnqh_x003D;25&y&tnqh_x003D;5*

*(accessed May 5, 2016) and the inclusion of snow events since scavenging by snow is more*

*efficient than by rain due to the larger surface area of snow (snow: 700 $cm^2/g$, rain: 60*

*$cm^2/g$) (Kerbrat et al., 2008)."*

**Comment 9.**

**Page 10, line 282:** "Therefore, all of the Hg deposited…" What fraction is lost? What fraction is retained? This could be really interesting.

**Response 9.**

Unfortunately we did not make any direct measurements of what was collected on the leaves and how much remained after a precipiation event so we can not address this question.

However we did add a bit more discussion on **Line 296 to Line 300**.

*"Some of the deposited Hg can be washed off by rainfall and reemitted as GEM to the*

*atmosphere (Jiskra et al., 2015; Rea et al., 2001). Therefore, all of the Hg deposited on the*

*foliar surfaces is not in the throughfall. Throughfall also incorporates GOM and PBM that is*

*adsorbed from the atmosphere by leave since GOM is soluble and it is likely readily washed*

*off during rain events (Blackwell and Driscoll, 2015)."*

**Comment 10.**

**Page 10, lines 294-296:** "… the rest of the variation is likely due to variations in local…"

The "rest" here being >80%, correct? A more rigorous explanation of the majority of the variability seen in the data would be helpful. Being able to explain less than 20% does not give confidence in the interpretation.

**Response 10.**

As suggested we added further explanations as follows on **Line 312 to Line 321**.

*"The rest of the variation is likely due to meteorological parameters that differ between*

*events, for example temperature and precipitation type (Gratz et al., 2009) and variations in*

*ambient Hg speciation and PBM particle size distributions due to differing impacts of local*

*and regional sources (Blackwell and Driscoll, 2015). There was a statistically significant*

*positive correlation between rainfall depth and TM deposition flux in precipitation ($r^2$ =*

*0.34) (p < 0.05), similar to what was found iiin previous studies (Choi et al., 2008; Gratz et*

*al., 2009; Maremoto and Matsuyama, 2014; Shanley et al., 2015; Wang et al., 2014),*

*suggesting that the TM deposition flux increased during large events even though continuous*

*rain diluted the TM mass. However, a large rainfall depth does not affect wet deposition*

*fluxes if atmospheric concentrations of GOM and PBM are low (Zhang et al., 2012)."*

**Comment 11.**

**Page 10, lines 307-309:** Why is an r2=0.59 (p<0.05) "significantly correlated" and r2=0.56 (p<0.05) "moderately correlated"?

**Response 11.**

Those should be the same - we corrected "moderately correlated" to "significantly correlated" as follows on **Line 330**.

" .... concentrations were also significantly correlated ($r^2$ = 0.56) (p < 0.05) (Table 1)"

**Comment 12.**

**Page 12:** If you are missing data in January and February, the stated assumption that "fluxes were assumed to be equal to the average of the flux of the month before" doesn't make sense. How are you handling consecutive months of missing data?

**Response 12.**

We corrected this mistake about missing data as follows on **Line 377 to Line 380**.

*"Due to sampler (Tekran 2537A) malfunctions in January, February and April, fluxes were assumed to be equal to the average of the flux of the previous and subsequent month. If only one month of data were available, it was assumed to be the same as the missing month"*

**Comment 13.**

**Page 13, lines 382-383:** It would be useful here to be specific and describe what "both input approaches" are. It's not that clear what approaches you mean.

**Response 13.**

As suggested, we added further explanations as follows on **Line 406 to Line 408**.

*"The yearly estimated budget of Hg was calculated using both input approaches (Total input = wet deposition + dry deposition or Total input = throughfall + litterfall) as follows..."*

**Comment 14.**

**Section 3.8:** This section is disjointed and lacks cohesion. Revision strongly encouraged, with a focus on building a logical progression.

**Response 14.**

As suggested this section has been revised as follows on **Line 406 to Line 419**.

*"The yearly estimated budget of Hg in this study site was calculated using both input approaches (Total input = wet deposition + dry deposition or Total input = throughfall + litterfall) as follows. 1) Input to the forest canopy (wet deposition in an open area: 4.3 μg m$^{-2}$ yr$^{-1}$ plus dry deposition in the forested area: 9.9 μg m$^{-2}$ yr$^{-1}$) minus output (emissions from soil 6.8 μg m$^{-2}$ yr$^{-1}$ plus accumulation in the soil 0.6 μg m$^{-2}$ yr$^{-1}$) resulting in a net flux of 6.8 μg m$^{-2}$ yr$^{-1}$. 2) The alternative method yields input (throughfall: 6.7 μg m$^{-2}$ yr$^{-1}$ plus litterfall: 4.6 μg m$^{-2}$ yr$^{-1}$) minus output (emissions from soil: 6.8 μg m$^{-2}$ yr$^{-1}$ plus accumulation in the soil: 0.6 μg m$^{-2}$ yr$^{-1}$) resulting in a net flux of 3.9 μg m$^{-2}$ yr$^{-1}$. For comparison at the Lehstenbach catchment in Germany, the estimated net fluxes were similar: 6.8 μg m$^{-2}$ yr$^{-1}$ (Schwesig and Matzner, 2000) and in the Experimental Lakes Area (ELA) watersheds in Canada, the flux was 3 ~ 4 μg m$^{-2}$ yr$^{-1}$ (St. Louis et al., 2001). However, for the Lake Langtjern spruce forest in southeast Norway (20.1 μg m$^{-2}$ yr$^{-1}$) (Larssen et al., 2008) and Huntington Wildlife forest (15.9 μg m$^{-2}$ yr$^{-1}$ in deciduous, 26.8 μg m$^{-2}$ yr$^{-1}$ in conifer) (Blackwell et al., 2014), the estimated fluxes were higher than in this study."*

**Comment 15.**

**Conclusions:** The manuscript needs a Conclusions section. Without Conclusions, the manuscript incomplete and doesn't seem mature enough for publication. A couple of strong synthesis statements from the authors about why their results add to our knowledge in the Hg field would really help the paper.

**Response 15.**

As suggested, we added a conclusions section as follows on **Line 423 to Line 451**.

*"Hg in dry and wet deposition, throughfall and litterfall and Hg volatilization from soil were measured from August 2008 to February 2010 to identify the factors influencing the amount of atmospheric Hg deposited to forested areas in a temperate deciduous forest in Korea. In*

*addition measured and theoretical dry deposition were compared. The GOM fluxes were low*

*in fall and increased towards the spring. PBM fluxes were lowest in fall and peaked in*

*summer. The estimated and directly measured deposition fluxes were not significantly*

*correlated likely due to loss of litter samples by wind or wash-off by rainfall and the fact that*

*accumulation in leaf stomata was not characterized in the direct dry deposition measurement*

*technique. The average VWM Hg concentration in throughfall was approximately 2.4 times*

*higher than in precipitation due to wash off of previously deposited Hg from the foliage. Both*

*were higher in winter due to increased concentrations in snow events relative to rain events*

*likely due to enhanced scavenging of GOM and PBM. TM in litterfall fluxes were highest in*

*fall when the leaves were dropped and lowest in summer from June to August. Hg emission*

*fluxes from soil resulted in a cumulative annual volatilization of 6.8 $\mu g$ $m^{-2}$ $yr^{-1}$ of GEM.*

*Based on this all data, the yearly accumulation of Hg in the deciduous forest was*

*calculated using two input approaches (total input = throughfall + litterfall or wet deposition*

*+ dry deposition and total output: emission from soil + TM in soil). Using this approach the*

*accumulation of Hg were 6.8 and 3.9 $\mu g$ $m^{-2}$ $yr^{-1}$ respectively.*

*There are several uncertainties associated with this study as discussed above. The*

*primary ones include that fact that dry deposition measured with the surrogate surface does*

*not account for accumulation in leaf stomata yet this technique yielded a larger flux than to*

*litterfall + throughfall – wet deposition. Litterfall can be lost from the sampler by wind and*

*Hg can be lost from the collected litter due to washoff from rainfall due to relatively long*

*sampling periods. The differences in the approaches suggest that approximately half of the*

*GEM stored in the leaf may be released to back to the atmosphere. DFCs can alter measured*

*fluxes because they cover the soil potentially blocking some UV light. In addition, several*

*months of measurements were missed. Finally grab samples for TM in soil may not capture*

*the true variability in the forest. Additional work should focus on better quantifying dry*

*deposition, TM in soil water, overflow rate and biogeochemical recycling within the forest*

*canopy and understory."*

**Comment 16.**

**Supporting Information:** I encourage the authors to make their data available in the SI. This will make it easier for other interested scientists, especially modelers, to compare against the data in Korea. The mercury community will be excited about this dataset and want to weave it into their comparisons – make it easy for them!

**Response 16.**

As suggested, we revised SI as follows on **Line 31 to Line 50**.

**Response to Referee 2:**

**Comment 1**

Abstract is written in the way with number reporting, there is no significant conclusions and any new discovery.

**Response 1**

As suggested, we revised the Abstract as follows on **Line 34 to Line 51**.

*"In this study, mercury (Hg) in dry and wet deposition, throughfall and litterfall, and Hg*

*volatilization from soil were measured from August 2008 to February 2010 to identify the*

*factors influencing the amount of atmospheric Hg deposited to forested areas in a temperate*

*deciduous forest in Korea. For this location there was no significant correlation between the*

*estimated dry deposition flux (litterfall + throughfall – wet deposition) (6.7 $\mu$g m$^{-2}$ yr$^{-1}$) and*

*directly measured dry deposition (9.9 $\mu$g m$^{-2}$ yr$^{-1}$) likely due primarily to Hg losses from the*

*litterfall collector. Dry deposition fluxes in cold seasons (fall and winter) were lower than in*

*warmer seasons (spring and summer). The volume-weighted mean (VWM) Hg concentrations*

*in both precipitation and throughfall were highest in winter likely due to increased*

*scavenging by snow events. Since Korea experiences abundant rainfall in summer, VWM Hg*

*concentrations in summer were lower than in other seasons. Litterfall fluxes were highest in*

*the late fall to early winter when leaves were dropped from the trees (September to*

*November). The cumulative annual Hg emission flux from soil was 6.8 $\mu$g m$^{-2}$ yr$^{-1}$. Based on*

*these data, the yearly deposition fluxes of Hg calculated using two input approaches*

*(throughfall + litterfall or wet deposition + dry deposition), were 6.8 and 3.9 $\mu$g m$^{-2}$ yr$^{-1}$*

*respectively. This is the first reported study which measured the amount of atmospheric Hg*

*deposited to forested areas in Korea and thus our results provide useful information related*

*to Hg fate and transport in this part of the world."*

**Comment 2.**

Line 69-71, the original papers were not cited and PBM is particle-bound Hg, how can it be adsorbed on PM? You could say oxidized Hg or GOM.

**Response 2**

As suggested, we revised text and reference as follows on **Line 70 to Line 71**.

"PBM is created by GEM or GOM adsorbing to a particle (Lai et al., 2011)."

**Comment 3.**

Line 72-74, re-write.

**Response 3**

As suggested, we revised as follows on Line **71 to Line 74**.

"...... Atmospheric PBM transport is significantly affected by its particle size distribution and may contribute to both wet and dry deposition (Lynam and Keeler, 2002). Wet and dry deposition of atmospheric Hg is an important input to the aquatic and terrestrial ecosystems"

**Comment 4.**

Line 84-86, not clear, also update the reference here Line 88-89

**Response 4**

As suggested, we revised and updated this text as follows on **Line 80 to Line 105**.

*"Dry deposition to leaves compromises a large proportion of litterfall (Grigal, 2002; St.*

*Louis et al., 2001). Previous investigations (Fu et al., 2009) estimated dry deposition to*

*forested areas as litterfall + throughfall – wet deposition. However, there are many variables*
*that can adversely influence this technique including reemitted Hg from beneath the canopy*
*and sampling artifacts.   Directly measuring dry deposition with a surrogate surface is an*
*alternative approach, although there is no universally accepted method on how to make these*
*measurements.*

*Hg deposited onto plant surfaces can be revolatilized, incorporated into tissue or*
*washed off by precipitation (which is deemed throughfall) which often results in throughfall*
*having higher Hg concentrations than precipitation (Iverfeldt, 1991; Kolka et al., 1999;*
*Munthe et al., 1995; Choi et al., 2008; Grigal et al., 2000; Schwesig and Matzner, 2000).*
*Litterfall is dead plant material such as leaves, bark, needles and twigs that has fallen to the*
*ground. Litterfall carries new Hg inputs from the atmosphere to the forest floor and also Hg*
*recycled from volatilization from soils and other surfaces. Throughfall and litterfall*
*contribute to the biochemical recycling of atmospheric Hg in forest systems (St. Louis et al.,*
*2001) and are important Hg inputs that result in Hg accumulation in forest systems*
*(Blackwell and Driscoll, 2015).*

*The deposition of Hg in the forest ecosystem is complicated because of complex*
*interactions between atmospheric Hg and the canopy, including oxidation of Hg on leaf*
*surfaces (Blackwell and Driscoll, 2015; Iverfeldt, 1991), deposition of GOM and PBM on*
*leaf surfaces (Blackwell et al., 2014; Blackwell and Driscoll, 2015; St. Louis et al., 2001),*
*stomatal uptake of atmospheric GEM (Fu et al., 2010; Iverfeldt, 1991; Lindberg et al., 1991;*
*St. Louis et al., 2001), root uptake of dissolved Hg in soil and soil water and stomatal uptake*
*of GEM that was volatilized from soils (Bishop et al., 1998; Cocking et al., 1995; Ma et al.,*
*2015; St. Louis et al., 2001). Recycled Hg would increase throughfall and litterfall*
*concentrations (St Louis. et al., 2001)."*

**Comment 5.**

Line 77-79, add Selin et al., 2007 and Lindberg et al., 2007

**Response 5**

As suggested, we added Selin et al., 2007 and Lindberg et al., 2007 as follows on **Line 77 to Line 79**.

*"…. , resulting in adverse health and ecological effects (Ma et al., 2013; Lindberg et al.,*

*2007; Rolfhus et al., 2003; Selin et al., 2007; Weiss-Penzias et al., 2016; Zhu et al., 2014)."*

**Comment 6.**

how does uptake via roots impact Hg deposition. Also stomatal uptake of $Hg^0$ emitted from soils?

**Response 6**

As suggested, we added information as follows on **Line 102 to Line 105**.

*" .... root uptake of dissolved Hg in soil and soil water and stomatal uptake of GEM that was*

*volatilized from soils (Bishop et al., 1998; Cocking et al., 1995; Ma et al., 2015; St. Louis et*

*al., 2001). Recycled Hg would increase throughfall and litterfall concentrations (St Louis. et*

*al., 2001)."*

**Comment 7.**

I don't understand this sentence. Line 133, please discuss problems from using KCl coated quartz surface. Lyman et al., 2010; Huang et al., 2013/2015, McClure et al., 2015, Lynam and

Keeler 2006 Sampling method, what are the time periods?

**Response 7**

As suggested, We revised and discussed problems from using KCl coated quartz surface as follows on **Line 141 to Line 154**.

*"Some studies have investigated using a surrogate surfaces to measure dry deposition*

*(Lyman et al., 2007; Peterson and Gustin, 2008). Surrogate surfaces is better control over*

*exposure times than those provided with natural vegetation (Lai et al., 2011). Surrogate*

*surfaces with cation exchange membranes could be collected very small aerosol fraction by*

*diffusion (Lyman et al., 2007; Huang and Gustin, 2015b). However, we collected direct dry*

*deposition using a surrogate surfaces with quartz filters. Similar to previous studies (Lai et*

*al., 2011; Yi et al., 1996), the dry deposition sampler was equipped with a knife-edge*

*surrogate surface (KSS) sampler with the collection media facing up. Forty seven-mm quartz*

*filters were used to measure PBM deposition and KCl-coated quartz filters were used to*

*measure GOM + PBM deposition. The quartz filter and KCl-coated quartz filter (soaked in*

*KCl solution for 12h and dried on clean bench) were pre-baked in a quartz container at 900*

*ºC for PBM and 525 ºC for GOM + PBM. Before weekly sampling, the filters were placed on*

*a filter holder base and held in place with a retaining ring and then were placed on the KSS.*

*Filters exposed to the atmosphere for approximately one week and two side-by-side samples*

*were deployed during each dry day."*

**Comment 8.**

Analytical method, did the author develop the thermal desorption method? If not please cite references.

      **Response 8**

As suggested, we added references as follows on **Line 187 to Line 188**.

*"....... zero air passed through until the Hg concentration was zero (Kim et al., 2009; Kim et*

*al., 2012)."*

**Comment 9.**

If I understand this correctly, KCl QFF was heated to 525C and QFF was heated to 900C to separate GOM and PBM. Two questions here. 1. Is dry deposition collected up facing or down facing? and how up/down facing impact measurement? 2. Is this possible for GOM

attach on QFF and quantified as PBM, and PBM attach on KCl-QFF and quantified as GOM?

      **Response 9**

As suggested, we added information about sampling method of dry deposition as follows on

**Line 141 to Line 154 and Line 192 to Line 193**.

*Question 1 :*

*Some studies have investigated the use of surrogate surfaces to directly measure Hg dry*

*deposition (Lyman et al., 2007; Peterson and Gustin, 2008). Surrogate surfaces allow better*

*control over exposure times than those provided with natural vegetation (Lai et al., 2011).*

*Surrogate surfaces with cation exchange membranes have been useful for measuring GOM however they may collect a very small aerosol fraction by diffusion (Lyman et al., 2007; Huang and Gustin, 2015b). Similar to previous studies (Lai et al., 2011; Yi et al., 1996), in this project the dry deposition sampler was equipped with a knife-edge surrogate surface (KSS) sampler with the collection media facing up. Forty seven-mm quartz filters were used to measure PBM deposition and KCl-coated quartz filters were used to measure GOM + PBM deposition. The quartz filter and KCl-coated quartz filter (soaked in KCl solution for 12h and dried on clean bench) were pre-baked in a quartz container at 900 ºC for PBM and 525 ºC for GOM + PBM. Before weekly sampling, the filters were placed on a filter holder base and held in place with a retaining ring and then were placed on the KSS. Filters exposed to the atmosphere from approximately one week and two side-by-side samples were deployed during each dry day.*

*Question 2 :*

*It was assumed that GOM deposition was equal to the flux measured by the KCl-coated quartz filter minus the flux measured by the quartz filter.*

**Comment 10.**

What is the recovery for the thermal desorption system? Recovery for Tekran 2537 direct injection 87% is too low usually from 93-107%. How many sampling time periods? Only 4 field blanks? Why? Volatilization from soil, what are MDL or blanks?

     **Response 10**

As suggested, we revised section 2.4.1 and 2.4.4 as follows on **Line 209 to Line 214 and Line 232 to Line 235**.

*"Automated daily calibration of Tekran 2537A routinely was performed using an internal permeation source. Two-point calibrations (zero and span) were performed separately for each pure gold cartridge. A recovery of $102 \pm 2.9\%$ ($r^2 > 0.9995$) ($n = 4$) was measured by directly injecting knowing amounts of five Hg standards which was connected to zero air. The Method Detection Limit (MDL) determined by measuring the Hg concentration in zero air was 0.04 ng m$^{-3}$. Additional information is described in the SI..................Before flux chamber measurements automated calibration was performed using the internal permeation*

*source connected to the Tekran 2537A and Tekran 1110 dual sampling unit. External*

*calibration and MDLs for this instrument are described above."*

**Comment 11.**

Section 3.1, if you only have a short time period during each season, how can you really see the seasonal pattern? Please add more detail information for sampling plan.

**Response 11**

As suggested, we added information about sampling plan as follows on **Line 241 to Line 242**.

*"Weekly sample were collected using quartz (PBM) and KCl coated quartz filters (GOM)"*

**Comment 12.**

What statistical test are you using, please add information for every place you mention significant difference.

**Response 12**

As suggested, we added information about significant difference as follows on **Line 247,**

**Line 265, Line 272, Line 290, Line 327, Line 396.**

*"Nonparametric Mann-Whitney U tests....."*

**Comment 13.**

Line 281, what is mechanical weathering?

**Response 13**

As suggested, we revised as follows on **Line 294 to Line 296**.

*"Other possible sources of Hg in throughfall are leaching and biogeochemical recycling of*

*Hg from foliage (St. Louis et al., 2001)."*

**Comment 14.**

Most references are also out of date.

**Response 14**

As suggested, we added recent references.

---

## Referee Report (RR1)

Review of Total Atmospheric Mercury Deposition in Forested Areas in Korea
By Han et al., 2016 ACP

In general, this paper has been improved significantly; however, I still have some questions and concerned. I suggest to accept this paper after a minor revision.

Line 35, monthly or weekly measurements?
Line 37, monthly data for the correlation? How many data point?
Line 49-51, how this would be useful for global mercury research?

Line 68-70, re-write the sentence, it is difficult to understand.
Line 70, "created" use another word, I suggest to use formed.
Line 106-107, references
Line 131, references, or probably delete it, Hg methylation is a very complicated process, it would be better to explain this in detail if possible.

Line 142, Huang et al., 2015 passive sampler review paper and 2011 Atmos. Env. wind tunnel tests.
The authors need to talk about the field blanks for dry/wet deposition and all other measurements
Line 179 DFCs were placed 2 cm under the soil? Re-write, I don't think this is possible, probably say "The bottom 2 cm of DFCs is covered by soil and soil surface to the chamber top is XX cm" some thing like this. The sentence sounds like the chamber is fully covered by soil.
Line 180, UV light needs some references
Line 192-193, explain what are the uncertainties here
Line 221, what is the RPD range? I expect that might be large, but it should be fine
What are the max capacity of these surface, in case you did not over load them?

Line 257-258, previous studies show no GEM collected on KCl surface, and in Zhang et al 2012, they discussed the potential GEM uptaken by dry deposition measurements is due to the usage of acidified BrCl. Since BrCl was not used in this study, this is not a suitable statement.
Line 262-265, 269-272, if you have figures or tables to present the data, you don't need to repeat the data again in text.
Line 276-278, could the author please do the analysis in detail? In the North American, we are seeing winter time low PBL, I agree GEM concentrations will increase, but I never see GOM concentrations increase in low PBL condition. The authors cited two papers here, Kim et al., 2009 and Seo et al., 2015, I went back to read these two papers, Seo et al., 2015 cited Kim et al., 2009 to make the statement, and Kim et al., 2009 cited Blanchard et al., 2002 to make the statement. None of Seo et al., 2015 and Kim et al., 2009 did a detail analysis on this. I just wonder could the authors do a detail analysis on how PBL decreasing impact atmospheric GOM concentrations?
Line 312-315, could the authors discuss this in detail, is there any information measured at these sites supporting this statement?
Line 320-321, do not understand
Line 379-380, what are the uncertainties?
Line 384-402, re-write this paragraph. There are some things I suggest the authors can look into.
Estimated dry deposition should less or equal to measured dry deposition due to no canopy

resistance for KCl surface, no re-emissions for KCl surface. Similar concept for wet + dry deposition and throughfull + litterfall should be considered. There is no (or very small) re-emissions for wet + dry deposition; therefore, the numbers are totally making sense to me. However, the authors did not explain this in detail.

Line 413-419, we know atmospheric GOM concentrations at this site are higher than the numbers measured in Huntington Wildlife forest. However, the net flux in HWF is higher than the number at this site. Does this mean that Hg soil emissions are in Korea way higher the numbers in HWF? If this is true, what could be the reasons?

---

## Author Response (AR3)

May 21, 2016

Dear Editor,

We appreciate the reviewers' suggestions which have considerably improved the manuscript (**acp-2016-7**). Enclosed are point-by-point responses to the reviewers. We hope that with these changes the manuscript will be suitable for publication in "**Atmospheric Chemistry**

**and Physics**"

Thank you very much.

Sincerely,

Seung-Muk Yi

Professor, Dept. of Environmental Health, Graduate School of Public Health

Seoul National University, 1 Gwanak-ro, Gwanak-gu, Seoul 151-742, South Korea

Telephone: (82) 2-880-2736, Fax: (82) 2-762-9105, E-mail: yiseung@snu.ac.kr

**Response to Referee's Reports**

● Journal: ACP

● Title: Total Atmospheric Mercury Deposition in Forested Areas in Korea

● Author(s): Jin-Su Han, Yong-Seok Seo, Moon-Kyung Kim, Thomas M. Holsen, Seung-Muk Yi

● MS No.: acp-2016-7

● MS Type: Research article

● Special Issue: Data collection, analysis and application of speciated atmospheric mercury

**Response to Referee 1:**

**Comment 1.**

Page 3, lines 58-60. Weiss-Penzias et al. (2016) and Zhu et al. (2014) are not appropriate references. The toxicity of MeHg and threat to ecosystems was established long before these papers.

       **Response 1.**

We revised references as follows on **Line 60 to Line 61**.

*" ........ bioaccumulate and biomagnify through the food chain after it is methylated (Lindqvist, 1991; Schroeder and Munthe, 1998)"*

**Comment 2.**

Page 4, lines 97-105. It would make sense to include the work of Risch et al. (2011) Environmental Pollution.

       **Reponse 2.**

We revised line 97-105 as follows on **Line 105 to Line 108**.

*".......that was volatilized from soils (Bishop et al., 1998; Cocking et al., 1995; Ma et al., 2015; St. Louis et al., 2001). Also, the Hg in forest canopies can be emitted and reemitted from beneath the canopy (Risch et al., 2012). The Hg mass in litterfall have orginated from a large portion of dry deposition (Risch et al., 2012; St. Louis et al., 2001)."*

**Comment 3.**

Page 10, lines 283-285. Sigler et al. (2009) found a similar effect from snow scavenging in the northeastern US. A brief comparison to their work is perhaps worth including.

       **Response 3.**

We added effect from snow scavenging refer to Sigler et al. (2009) as follows on **Line 283 to Line 284**.

*"While, Sigler et al. (2009) reported that GOM is scavenged less efficiently during snow events."*

**Response to Referee 2:**

**Comment 1.**

Line 35, monthly or weekly measurements?

      **Response 1.**

We added measurement cycle as follows on **Line 34 to Line 36**.

*"In this study, mercury (Hg) was sampled weekly in dry and wet deposition and throughfall and monthly in litterfall, and as it was volatilized from soil from August 2008 to February 2010...."*

**Comment 2.**

Line 37, monthly data for the correlation? How many data point?

      **Response 2.**

We added as follows on **Line 37 to Line 38**.

*"For this location there was no significant correlation between the estimated monthly dry deposition flux...."*

**Comment 3.**

Line 49-51, how this would be useful for global mercury research?

      **Response 3**

We added some information as follows on **Line 50 to Line 52**.

*"... and thus our results provide useful information to compare against data related to Hg fate and transport in this part of the world."*

**Comment 4.**

Line 68-70, re-write the sentence, it is difficult to understand.

      **Response 4.**

We re-wrote the the sentence as follows on **Line 70 to Line 71**.

*"The dry deposition velocity is similar to $HNO_3$ ($1\sim5$ cm sec$^{-1}$) if it is assumed that all GOM is in the form of $HgCl_2$ (Petersen et al., 1995)"*

**Comment 5.**

Line 70, "created" use another word, I suggest to use formed.

**Response 5.**

We revised from "created" to "formed" as follows on **Line 71**.

*"PBM is formed by GEM or GOM...."*

**Comment 6.**

Line 106-107, references

**Response 6.**

We added some references as follows on **Line 109 to Line 110**.

*"To date there have been few studies (Blackwell et al., 2014; Choi et al., 2008; Rea et al., 2001) that have estimated atmospheric Hg deposition to forested areas and none in Korea."*

**Comment 7.**

Line 131, references, or probably delete it, Hg methylation is a very complicated process, it would be better to explain this in detail if possible.

**Response 7.**

We deleted about Hg methylation as follows on **Line 133 to Line 134**.

*"This area contains rivers, a flood plain, agricultural land, residential areas, forests, and wetlands. Therefore, the study sites are....."*

**Comment 8.**

Line 142, Huang et al., 2015 passive sampler review paper and 2011 Atmos. Env. wind tunnel tests.

**Comment 8.**

We added some information about surrogate surfaces refer to Huang et al., 2011 as follows on **Line 144 to Line 149**.

*"Surrogate surfaces allow better control over exposure times than those provided with*

*natural vegetation (Lai et al., 2011). However, surrogate surfaces, being smooth, may not*

*mimic Hg dry deposition to natural rougher surfaces (Huang et al., 2011). Surrogate surfaces*

*with cation exchange membranes have been useful for measuring GOM however they may*

*collect a very small aerosol fraction by diffusion (Huang and Gustin, 2015; Lyman et al.,*

*2007).*

**Comment 9.**

The authors need to talk about the field blanks for dry/wet deposition and all other measurements

**Comment 9.**

We mentioned field blank for dry/wet deposition in the SI as follows on **Line 35 to Line 42**.

*".... Field blank for GOM (n = 51) and PBM (n = 46) were 0.21 and 0.19 ng $m^{-2}$ $hr^{-1}$*

*respectively..................... Field blanks were collected monthly from September to December*

*and yielded Hg concentrations of 0.36 ± 0.05 ng $L^{-1}$.*

**Comment 10.**

Line 179, DFCs were placed 2 cm under the soil? Re-write, I don't think this is possible, probably say "The bottom 2 cm of DFCs is covered by soil and soil surface to the chamber top is XX cm" something like this. The sentence sounds like the chamber is fully covered by soil.

**Response 10.**

We revised these sentence as follows on **Line 183**.

*"The bottom 2 cm of DFCs (3.78L) were covered by soil."*

**Comment 11.**

Line 180, UV light needs some references

**Response 11.**

We added some references as follows on **Line 184 to Line 185**.

*"The DFCs were made of glass and polycarbonate which may block some UV light (Choi and Holsen, 2009; Skinner, 1998)."*

**Comment 12.**

Line 192-193, explain what are the uncertainties here

**Response 12.**

We explained about uncertainties as follows on **Line 197 to Line 199**.

*".....measured by the KCl-coated quartz filter minus the flux measured by the quartz filter. However, recent studies (Lyman et al., 2010) reported potential sampling artifacts in the presence of $O_3$. "*

**Comment 13.**

Line 221, what is the RPD range? I expect that might be large, but it should be fine

What are the max capacity of these surface, in case you did not over load them?

**Response 13.**

We already mentioned RPD range in the SI as follows on **Line 41**.

*".....respectively with an RPD of 3 ~ 13%."*

**Comment 14.**

Line 257-258, previous studies show no GEM collected on KCl surface, and in Zhang et al 2012, they discussed the potential GEM uptaken by dry deposition measurements is due to the usage of acidified BrCl. Since BrCl was not used in this study, this is not a suitable statement.

**Response 14.**

We deleted this statement.

*"This suggests that GEM may contribute to the measured dry deposition (Zhang et al., 2012)"*

**Comment 15.**

Line 262-265, 269-272, if you have figures or tables to present the data, you don't need to repeat the data again in text.

**Response 15.**

We revised Section 3.2 as follows on **Line 267 to Line 272**.

*"The average VWM concentration in precipitation (n = 35) and throughfall (n = 44) are*

*shown Fig.3. Nonparametric Mann-Whitney U tests indicated that there were no statistically*

*significant differences in the VWM TM concentration between winter 2009 and other seasons*

*which is probably related with the small number of samples. The VWM TM concentration in*

*winter 2009 was statistically significantly higher than fall 2009 (p = 0.007), spring 2009 (p =*

*0.035), and summer 2009 (p = 0.001) in throughfall."*

**Comment 16.**

Line 276-278, could the author please do the analysis in detail? In the North American, we are seeing winter time low PBL, I agree GEM concentrations will increase, but I never see

GOM concentrations increase in low PBL condition. The authors cited two papers here, Kim et al., 2009 and Seo et al., 2015, I went back to read these two papers, Seo et al., 2015 cited

Kim et al., 2009 to make the statement, and Kim et al., 2009 cited Blanchard et al., 2002 to make the statement. None of Seo et al., 2015 and Kim et al., 2009 did a detail analysis on this.

I just wonder could the authors do a detail analysis on how PBL decreasing impact atmospheric GOM concentrations?

**Response 16.**

Unfortunately we are not aware of a simple way to calculate the average PBL height by season. We revised reference and paragraph as follows on Line **273 to Line 275**.

*"The high VWM Hg concentrations in precipitation and throughfall in winter were likely*

*associated with the combined effects of reduced mixing heights (Blanchard et al.,*

*2002)........."*

**Comment 17.**

Line 312-315, could the authors discuss this in detail, is there any information measured at these sites supporting this statement?

**Response 17.**

We added some information measured at these sites as follows on **Line 311 to Line 315**.

*"The rest of the variation is likely due to meteorological parameters that differ between*

*events (Gratz et al., 2009), for example temperature(Table S3) and precipitation type (Rain,*

*Snow, Mixed) and variations in ambient Hg speciation and PBM particle size distributions*

*due to differing impacts of local and regional sources (Blackwell and Driscoll, 2015)."*

**Comment 18.**

Line 320-321, do not understand

**Response 18.**

We revised this paragraph as follows on **Line 319 to Line 320**.

*"However, a large rainfall depth does not affect wet deposition fluxes significantly if GOM*

*and PBM concentrations are low (Zhang et al., 2012)"*

**Comment 19.**

Line 379-380, what are the uncertainties?

**Response 19.**

We mentioned the uncertainties as follows on **Line 375 to Line 380**.

*"The cumulative annual Hg emission flux was 6.8 $\mu g\ m^{-2}\ yr^{-1}$ (Fig. 5). Due to sampler*

*(Tekran 2537A) malfunctions in January, February and April, fluxes were assumed to be*

*equal to the average of the flux of the previous and subsequent month. If only one month of*

*data were available, it was assumed to be the same as the missing month. For comparison*

*the annual Hg emission flux would be 4.8 $\mu g\ m^{-2}\ y^{-1}$ if only measured data were used."*

**Comment 20.**

Line 384-402, re-write this paragraph. There are some things I suggest the authors can look into. Estimated dry deposition should less or equal to measured dry deposition due to no canopy resistance for KCl surface, no re-emissions for KCl surface. Similar concept for wet +

dry deposition and throughfull + litterfall should be considered. There is no (or very small)

reemissions for wet + dry deposition; therefore, the numbers are totally making sense to me.

However, the authors did not explain this in detail.

**Response 20.**

We added some texts as follows on **Line 384 to Line 407**.

*"Fu et al. (2009) estimated dry deposition to be equal to litterfall + throughfall – wet*

*deposition. Using the data presented here, the estimated dry deposition flux (6.7 $\mu$g m$^{-2}$ yr$^{-1}$)*

*was lower than measured dry deposition (9.9 $\mu$g m$^{-2}$ yr$^{-1}$) and there was no significant*

*correlation between the two methods ($r^2 = 0.22$) ($p = 0.65$). One of the reasons for the*

*directly measured flux to be larger than the estimated flux is likely because there is no*

*canopy resistance for, or re-emission from, the KCl coated surrogate surface. The differences*

*in the estimates could be due to the loss of litter samples by wind or Hg losses from the*

*collected litter due to meteorological conditions such as rainfall (Blackwell et al., 2014) due*

*to relatively long sampling periods (1 month). However dry deposition collected with a*

*surrogate surface doesn't include accumulation in leaf stomata which may underestimate dry*

*deposition using this technique and since it is a smooth surface may collect less deposition*

*than a rougher surface.*

*The annual input flux calculated by summing wet deposition plus measured dry*

*deposition (14.3 $\mu$g m$^{-2}$ yr$^{-1}$) was higher than the input flux calculated by summing*

*throughfall + litterfall (12.8 $\mu$g m$^{-2}$ yr$^{-1}$) (Fig. 6). This difference is likely, at least in part, due*

*to the fact that no Hg is reemitted from wet and dry deposition as happens for litterfall.*

*Nonparametric Mann-Whitney U tests indicated that there were not statistically significant*

*differences ($r^2 = 0.14$) ($p = 0.98$). In general, wet + dry deposition was larger than*

*throughfall plus litterfall except during fall when leaves were being actively dropped from the*

*trees. The largest difference was in July during a period of significant precipitation (about*

*26.3 % of the total amount in 2009). This difference is most likely due to the many reactions*

*and transformations on the leaf surface that aren't mimicked with the surrogate surface*

*including re-emission (Rea et al., 2001)."*

**Comment 21.**

Line 413-419, we know atmospheric GOM concentrations at this site are higher than the numbers measured in Huntington Wildlife forest. However, the net flux in HWF is higher than the number at this site. Does this mean that Hg soil emissions are in Korea way higher the numbers in HWF? If this is true, what could be the reasons?

**Response 21.**

*"Measured Hg soil emission fluxes in this study site (4.8 $\mu g$ $m^{-2}$ $yr^{-1}$) were lower than HWF*

*(7.0 $\mu g$ $m^{-2}$ $yr^{-1}$)."*

[revised manuscript text omitted]